# Cyclic 5-membered disulfides are not selective substrates of thioredoxin reductase, but are opened nonspecifically

Jan G. Felber [1], Lena Poczka[1], Karoline C. Scholzen [2], Lukas Zeisel [1], Martin S. Maier[1], Sander Busker[2,6], Ulrike Theisen [3], Christina Brandstädter[4], Katja Becker[4], Elias S. J. Arnér [2,5], Julia Thorn-Seshold [1] & Oliver Thorn-Seshold [1]✉

The cyclic five-membered disulfide 1,2-dithiolane has been widely used in chemical biology and in redox probes. Contradictory reports have described it either as nonspecifically reduced in cells, or else as a highly specific substrate for thioredoxin reductase (TrxR). Here we show that 1,2-dithiolane probes, such as "TRFS" probes, are nonspecifically reduced by thiol reductants and redox-active proteins, and their cellular performance is barely affected by TrxR inhibition or knockout. Therefore, results of cellular imaging or inhibitor screening using 1,2-dithiolanes should not be interpreted as reflecting TrxR activity, and previous studies may need re-evaluation. To understand 1,2-dithiolanes' complex behaviour, probe localisation, environment-dependent fluorescence, reduction-independent ring-opening polymerisation, and thiol-dependent cellular uptake must all be considered; particular caution is needed when co-applying thiophilic inhibitors. We present a general approach controlling against assay misinterpretation with reducible probes, to ensure future TrxR-targeted designs are robustly evaluated for selectivity, and to better orient future research.

[1] Department of Pharmacy, Ludwig-Maximilians University Munich, Butenandtstr. 5-13, 81377 Munich, Germany. [2] Department of Medical Biochemistry, Karolinska Institutet, Solnavägen 9, 171 77 Stockholm, Sweden. [3] Zoological Institute, Cellular and Molecular Neurobiology, TU Braunschweig, Spielmannstr. 7, 38106 Braunschweig, Germany. [4] Interdisciplinary Research Centre (IFZ), Justus-Liebig University Giessen, Heinrich-Buff-Ring 26-32, 35392 Giessen, Germany. [5] Department of Selenoprotein Research, National Institute of Oncology, 1122 Budapest, Hungary. [6]Present address: Pelago Bioscience AB, 171 48 Solna, Sweden. ✉email: oliver.thorn-seshold@cup.lmu.de

Specific dithiol/disulfide-exchange reactions underlie a great number of crucial pathways in biology. Often, these are coordinated through conserved, highly specialised networks of oxidoreductases[1]. The thioredoxin reductase–thioredoxin (TrxR–Trx) system, and the glutathione reductase–glutathione–glutaredoxin (GR–GSH–Grx) system, are central "nodes" in these networks. TrxR (nM cellular concentration) passes reducing equivalents from NADPH to Trx-fold effector proteins (µM). Similarly, GR (nM) passes reducing equivalents from NADPH to the redox-active peptide GSH (mM), that can directly function as a cellular reductant or be further shuttled to effector Grx proteins (µM). These systems drive hundreds of redox reactions vital to cellular metabolism, and also regulate protein activity, protein–protein interactions, and protein localisation by reversible dithiol/disulfide-type reactions[2]. Their complex homeostasis[3] is dysregulated in many diseases, particularly in autoimmune disorders and cancer[4], making Trx and TrxR promising therapeutic targets[5]. Designing selective probes or substrates that report on their activities or target these redox nodes, would enable a broad range of applications in both basic biological and applied biomedical research, and is, therefore, a subject of intense development both through genetic engineering and chemical biology approaches[6,7].

Disulfides are the typical native substrates of these redox systems, and both linear and cyclic disulfides have been exploited as artificial substrates in biophysics, materials chemistry and chemical biology. Driven by the high intracellular concentration of thiols (ca. 50 mM total, ca. 5 mM GSH) compared to low concentrations in blood or in the extracellular space, linear disulfides undergo irreversible and nonspecific thiol-disulfide interchange and reduction in cells (Fig. 1a)[8,9]. Linear disulfides are thus used for nonspecific intracellular release and/or activation of appended cargos, exploiting the cellular thiol pool.

By contrast, cyclic disulfides can exhibit very different kinetics and thermodynamics of thiol-disulfide interchange or reduction,

so they may have different specificities. Cyclic disulfides are found in nature, perhaps most remarkably in the epidithiodiketopiperazine class of natural products (ETPs)[10–12]. ETPs such as gliotoxin (Fig. 1b) and chaetocin feature a near-planar diketopiperazine that is 1,4-bridged by a disulfide with a CSSC dihedral angle of 0°, which is high in energy compared to unstrained linear disulfides (90°) or 6-membered alicyclic disulfides (60°). ETPs were initially reported to inhibit a range of enzymes and cause a variety of cellular effects, but these poorly reproducible bioactivities are now understood as nonspecific reactivity of their highly strained disulfide[13].

A particularly important cyclic disulfide which remains less clearly analysed is the 5-membered 1,2-dithiolane (Fig. 1c). This motif underlies the key cellular redox cofactor lipoic acid; it is also found in several natural products[14,15], and it has emerged as a motif of general interest within the last decade[16–18]. The strained 1,2-dithiolane is kinetically labile to thiol-disulfide interchange[19], which likely underpins its role as a redox cofactor. The disulfide's opening/reduction kinetics have made it the focus of numerous chemical biology approaches, although these have been predicated on two mutually contradictory views of its cellular behaviour: which we aim to examine and resolve in this paper.

Following one view, 1,2-dithiolane has been cast as an easily, nonspecifically, and irreversibly opened and/or reduced motif[20]. Whitesides' systematic disulfide investigations highlighted that its strained CSSC dihedral angle of ca. 30° destabilises it by more than 8 kJ/mol relative to linear disulfides, and that its reduction potential (ca. −240 to −270 mV) is not significantly below that of linear disulfides (ca. −230 mV). In the 1950s, Fava observed that 1,2-dithiolane undergoes thiol-disulfide interchange with alkyl thiolates ca. 5000 times faster than do linear aliphatic disulfides[21]. Creighton reported that its reduction by the vicinal dithiol dithiothreitol (DTT) is over 100 times faster still[22], and Whitesides showed that this rate is only 100-fold slower than the diffusion limit in DMSO[23]. With favourable thermodynamics and

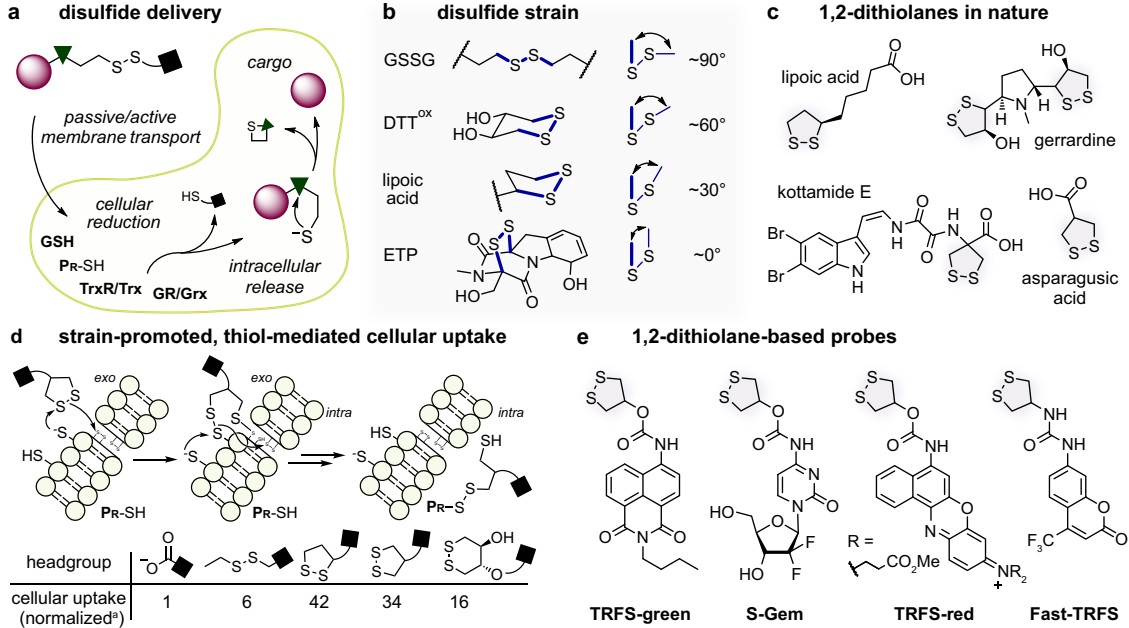

**Fig. 1 1,2-dithiolanes in chemical biology. a** Linear disulfide-based cellular delivery: irreversible cleavage of linear disulfides after cell entry leads to intracellular cargo release (**P$_R$-SH**: intracellular protein thiol). **b** From linear, to cyclic 6-membered, to increasingly ring-strained 5-membered disulfides, to ETPs. **c** Strained 1,2-dithiolanes in natural products. **d** Fast, irreversible and nonspecific thiol-disulfide interchange of 1,2-dithiolanes by exofacial thiols, followed by dynamic transmembrane exchange cascades, enhances cellular uptake and intracellular delivery. [ª:data from Matile et al.[30]]. **e** Some of the 1,2-dithiolane-based "TRFS" probes and prodrugs that have been reported as selective cellular substrates of thioredoxin reductase [TRFS = "Thioredoxin Reductase Fluorogenic Substrate"[16,18,45]].

high kinetic lability, 1,2-dithiolanes readily polymerise by nucleophile-catalysed ring-opening polymerisation, in particular in the presence of thiols[24,25]. Matile showed that the sterically less shielded 1,2-dithiolanes in asparagusic acid derivatives (two primary thiols) polymerise even more easily than do lipoic acid derivatives[26], giving them valuable applications e.g. in "SOSIPs" exploiting proximity-induced-polymerisation[27].

The intrinsic lability of 1,2-dithiolanes has also been extensively applied for thiol-mediated cellular uptake systems. Promoted by strain, 1,2-dithiolanes are attacked by cell surface thiols, then are internalised through a series of dynamic covalent reactions first with membrane thiols and later with intracellular thiols[28]. Attaching these strained disulfides to a molecular cargo thus drastically enhances its cellular uptake rate (Fig. 1d; overview in Supplementary Fig. 2a)[26,29–35]. It will be important to note that this process may be strongly affected by treatment with thiol-reactive species: cellular uptake rates are decreased many-fold by thiol-reactive electrophiles or oxidants; or instead enhanced by reducing agents[30,36,37]. Studying the role of membrane thiols in this process, and their connection to the intracellular redox environment, is also an expanding topic[38–42]. Other recent studies further demonstrate that 1,2-dithiolanes undergo fast strain-driven cross-linking or polymerisation, initiated either by thiols or by other nucleophiles[43,44].

In contrast however, 1,2-dithiolanes have also been reported as reduction-sensing units with a remarkable selectivity for TrxR. The fluorogenic TRFS-probes[16,18,45,46] and prodrugs[17] have been widely used for cellular studies, commercialised, and reviewed[47–51] (Fig. 1e; overview in Supplementary Fig. 2b). These probes have since been used in biology to study the role of TrxR in Parkinson's disease[52] and stroke[46], and have been employed for mechanistic validation of putative TrxR inhibitors during cellular screening approaches[18].

Which is the real situation? 1,2-dithiolane cannot be both highly and nonspecifically reactive, yet also TrxR-selective in the cellular setting. To develop a systematic understanding of redox biology, it is necessary to clarify such fundamental disagreements, and reveal why such contradictory interpretations could arise.

Towards this goal, we have investigated the environment-insensitive 1,2-dithiolane-based reduction-sensing probe **SS50-PQ** and systematically compared its performance to the prominent 1,2-dithiolane-based compound **TRFS-green**. We show that (a) 5-membered cyclic disulfides employed in these probes are nonspecifically reduced by a broad range of monothiols, dithiols, proteins and enzymes, and that (b) their cellular fluorogenicity is substantially independent of TrxR. We also show that (c) 1,2-dithiolane-based compound **Fast-T RFS** is fluorogenic even without cells and without TrxR, and that its fluorogenicity is consistent with reduction-independent, hydrophobicity-driven partitioning followed by catalytic ring-opening polymerisation. In the year after the preprint of this paper was posted, both **TRFS-green** and **Fast-TRFS** were re-evaluated by the same authors, now finding that they do indeed react at least with Grx, Trx, and GSH[53]. These revised reports are coherent with the much stronger conclusions we derive in this paper: that not only the **TRFS** probes, but indeed any 1,2-dithiolane probes, cannot be simplistically interpreted as cellular reporters for TrxR, Trx, Grx, GSH, or any combination of such species: the highly strained disulfide makes cellular readouts essentially not interpretable, and the compounds should not be used as probes in this manner.

Taken together, we and increasingly others are concluding that 1,2-dithiolane-based compounds cannot be selective cellular probes of TrxR. We signal to the community that previous studies interpreting the cellular performance of TRFS probes as TrxR reporters[16–18,45,46,52] will benefit from stringent re-evaluation, and the interpretations will probably be found to be wrong. We also signal to the community that the cellular performance of strained disulfide probes are probably best understood as nonspecifically monitoring thiol-mediated uptake rates, with the caveat that they are cellularly activatable by such a broad range of species and even non-reductive processes, that to interpret their cellular behaviour in terms of specific actors is also likely to be wrong. This does not prevent them from being legitimately applicable for enhanced delivery and nonspecific activation of trigger-cargo-systems, which opens up promising avenues for chemical biology. We also outline a strategy to control against assay misinterpretations with reducible probes, to promote progress towards a robust and useful toolset of probes for redox biology.

## Results

**Probe design**. We aimed to explore the properties of 1,2-dithiolanes using an environment-independent reduction-activated probe. Reduction-activated probes and prodrugs are typically trigger-cargo designs, where a reduction-sensitive trigger is connected to the cargo while masking a key functional group. Trigger reduction then results in a fragmentation reaction that restores activity by unmasking that key functional group. This concept has been used for a range of imaging agent[54,55] and drug[56–58] cargos.

The TRFS probes are also designed as trigger-cargo constructs with 1,2-dithiolane as the redox sensor, attached to aniline fluorophores through a carbamate (**TRFS-red**, **TRFS-green**; Fig. 1e) or urea (**Fast-TRFS**; Fig. 1e). For the carbamate probes, disulfide reduction and thiol cyclisation slowly releases the active aniline fluorophore; though the urea probe operates without aniline unmasking, and its fluorescence turn-on mechanism is not fully understood. Generally, most trigger-cargo disulfide probes employ aniline rather than phenol cargos (Supplementary Fig. 2b). While aniline carbamates have high hydrolytic stability[18], phenol-releasing designs would in many ways be more attractive targets, due to a large scope of potential cargos and to their improved release kinetics. Therefore, to test the reduction selectivity of 1,2-dithiolane with a rapidly-responding probe—and at the same time to establish a modular design that allows delivering a wide range of agents in the future—we created the phenol-releasing probe **SS50-PQ** (Fig. 2a).

The **SS50-PQ** design has several advantages. As a tertiary carbamate, this probe cannot decompose by $E_{1cB}$ elimination[59,60], avoiding the instability[18] that has blocked previous phenol-releasing 1,2-dithiolane probes. The choice of a 2-(2'-hydroxyphenyl)-4(3H)-quinazolinone (**PQ-OH**) as the cargo, ensures a fully off-to-on signal readout for carbamate cleavage. This is because only **PQ-OH**, but not **SS50-PQ**, can exhibit large-Stokes-shift ESIPT-based fluorescence due to intramolecular transfer of the phenolic hydrogen (ex/em 360/530 nm). Mechanistically, **PQ-OH** is released after cyclisation of the thiolate resulting from thiol-disulfide interchange or reduction expels the electron-poor phenolate (Fig. 2a); **PQ-OH** then precipitates upon reaching its low aqueous solubility limit (ca. 0.5 μM) activating its ESIPT fluorescence that is only visible in the solid state[61,62]. Therefore, fluorescence is unambiguously due to cyclisation-mediated cargo release; and as a solid-state fluorophore, it is not subject to environment-dependent effects. With the additional benefit of its large Stokes shift, this fully off-to-on system gives excellent signal-to-background fluorescence ratios of typically >100 without needing background subtraction, making it a sensitive and easily interpreted sensor of dithiolane cleavage.

**Probe synthesis**. 1,2-dithiolane **6** was prepared using an approach initially reported by Raines[63] followed by *N*-

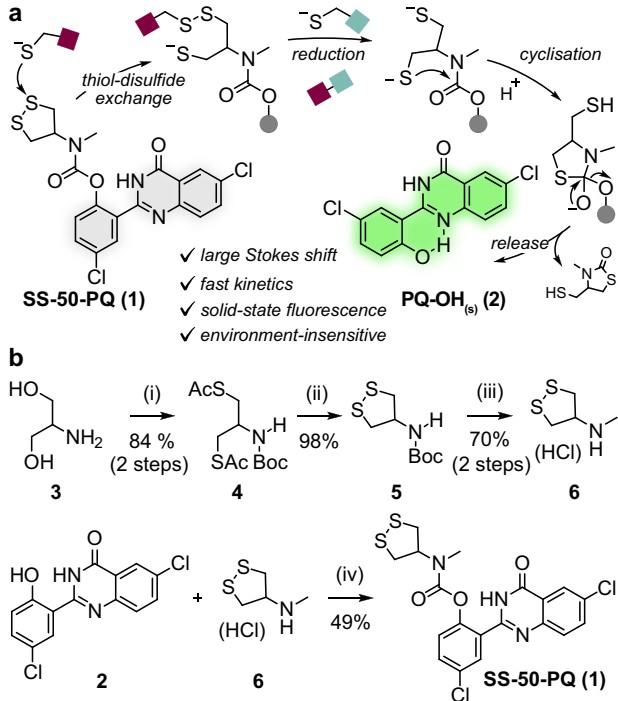

**Fig. 2 1,2-dithiolane probe design and synthesis. a** After opening or reduction of the 1,2-dithiolane in **SS50-PQ**, thiolate cyclisation releases the precipitating phenol **PQ-OH** that gives ESIPT-based fluorescence in the solid state. **b** Synthesis of **SS50-PQ**: (i) Boc$_2$O, NEt$_3$, dioxane/H$_2$O, r.t., 15 h (96%); *then* either MsCl, py, DCM *followed by* KSAc, acetone, 60 °C, 2 h (89%), *or* HSAc, PPh$_3$, DIAD, THF, 0 °C to r.t., 15 h (88%). (ii) KOH, MeOH, open to air, r.t., 15 h (98%). (iii) MeI, NaH, DMF, 0 °C to r.t., 0.5 h (70%). (iv) **PQ-OH**, triphosgene, NEt$_3$, DCM, 0 °C to r.t., 1 h; *then* **6**, NEt$_3$, DCM, 0 °C to r.t., 1 h (49%).

methylation and Boc-deprotection (Fig. 2b). All intermediates containing the 5-membered cyclic disulfide showed degradation upon standing, presumably producing linear polydisulfide oligomers (see Supplementary Information). This occured even without clear stimuli (e.g., stirring in dichloromethane). The cyclic monomer could typically be recovered by stirring in dilute solution, although isolating the monomer from this solution while avoiding re-polymerisation was not straightforward. The *N*-methylation step suffered particularly from polymerisation, until we found that the monomeric product could be extracted from methanol by hexane washes (see Supplementary Information). The final fluorogenic probe **SS50-PQ** was assembled by carbamate coupling with **PQ-OH**.

**1,2-dithiolane is unstable in probes.** Polymerisation was also observed for probe stock solutions in DMSO. Their maximal fluorescence, determined in a standardised reducibility test (aqueous buffer, pH 7.4, 10 eq of the quantitative disulfide reductant tris(carboxyethyl) phosphine, TCEP) decreased over time. We understood this as a consequence of polymerisation, since the hydrophobic, polymeric degradation products would have decreased accessibility to solvated reductants. Fresh probe stocks were, therefore, prepared immediately for each assay from powdered solid, then assayed for quality by comparison of the maximum TCEP-driven fluorescence to calibrations established by precipitating the theoretical amount of **PQ-OH**. Only stocks yielding TCEP-driven signals within 10% of the calibration intensity were used in assays. In fact, we re-prepared **SS50-PQ** five times during this research to maintain high-quality stocks.

**1,2-dithiolane is nonspecifically reduced by various thiols.** A disulfide trigger can only be enzyme-selective in the cellular context, if it resists signal generation from thiol-disulfide interchange or reduction by the cellular monothiol background (ca. 50 mM, of which ca. 5 mM GSH[64–66]). Hence, we began testing the potential for selectivity by performing cell-free incubations of **SS50-PQ** (10 μM) with GSH.

Probe "sensitivity or resistance" to challenge by a species, is often reported based on measurement at a single challenge concentration at a single timepoint. However, this simplification can misrepresent the situation by "hiding" signal that increases after a certain lag time. To provide a characterisation of probe resistance to monothiols, we titrated GSH over a wide concentration range (0.01 to 10 mM) and collected timecourse fluorescence data (Fig. 3a). For meaningful representation, we normalised the signals at each timepoint against the maximum possible fluorescence value at that timepoint (from the TCEP control: see Fig. 3a). This normalisation is important because it separates the upstream kinetics of reduction, from the potentially slower downstream kinetics of fragmentation which otherwise can obscure sensitivity to reduction; so it allows direct comparison of experiments relative to their theoretical maxima, and can be generally recommended. We also compared dose-response curves from various endpoint times (Supplementary Fig. 3) to ensure that any presented curve is representative of the probe's general behaviour.

We observed strong, fast probe response to even subphysiological GSH levels. The GSH concentrations causing half-maximal fluorescence ("EC$_{50}$$^{GSH}$") were ≲1 mM (Fig. 3a; Supplementary Fig. 3). This indicates that 1,2-dithiolane probes can be rapidly and fully reduced by cellular GSH concentrations, even without enzyme catalysis involved.

We also screened other monothiol reductants, e.g., cysteine (Cys), *N*-acetylcysteine (NAC), *N,N*-dimethyl-cysteamine (MEDA), and cysteamine (CA) and found fast probe activation (Fig. 3b) with similar concentrations and kinetics compared to GSH. This suggests that 1,2-dithiolane is generally instable to monothiols, so that probes derived from it will be rapidly activated by the intracellular thiol background. Matching expectations from Creighton[22] and Whitesides[23], the probe was quantitatively and rapidly triggered by equimolar amounts of vicinal dithiol DTT. We controlled for release by mechanisms other than interchange/reduction-triggered cyclisation, using serine (Ser) and glutathione disulfide (GSSG). The probe was entirely stable to non-reductive degradation by e.g. aminolysis, highlighting the stability of the tertiary phenolic carbamate, and supporting that interchange/reduction is its pathway for signal generation (Fig. 3b).

In summary, these assays show that 1,2-dithiolanes do not resist uncatalysed thiol-disulfide interchange and/or reduction even by monothiols and even at subphysiological concentrations, making them unlikely to be enzyme-selective in the cellular context.

**1,2-dithiolanes are nonspecifically reduced by redox-active proteins and enzymes.** We next tested probe reduction by redox proteins from the Trx/TrxR and Grx/GSH/GR systems. Each protein has multiple isoforms, as has been excellently reviewed[67]. We employed recombinant human Trx1 and Trx2; the thioredoxin-related protein TRP14, which features a vicinal dithiol/disulfide redox-active site that is similarly recovered by TrxR1; the oxidoreductases TrxR1, TrxR2 and GR; and human vicinal dithiol glutaredoxins Grx1 and Grx2. Both Trxs and Grxs have orders of magnitude higher cellular concentrations (ca. 10 μM) than their upstream TrxR and GR partners (ca. 20 nM), so we reflected these concentrations in our assays.

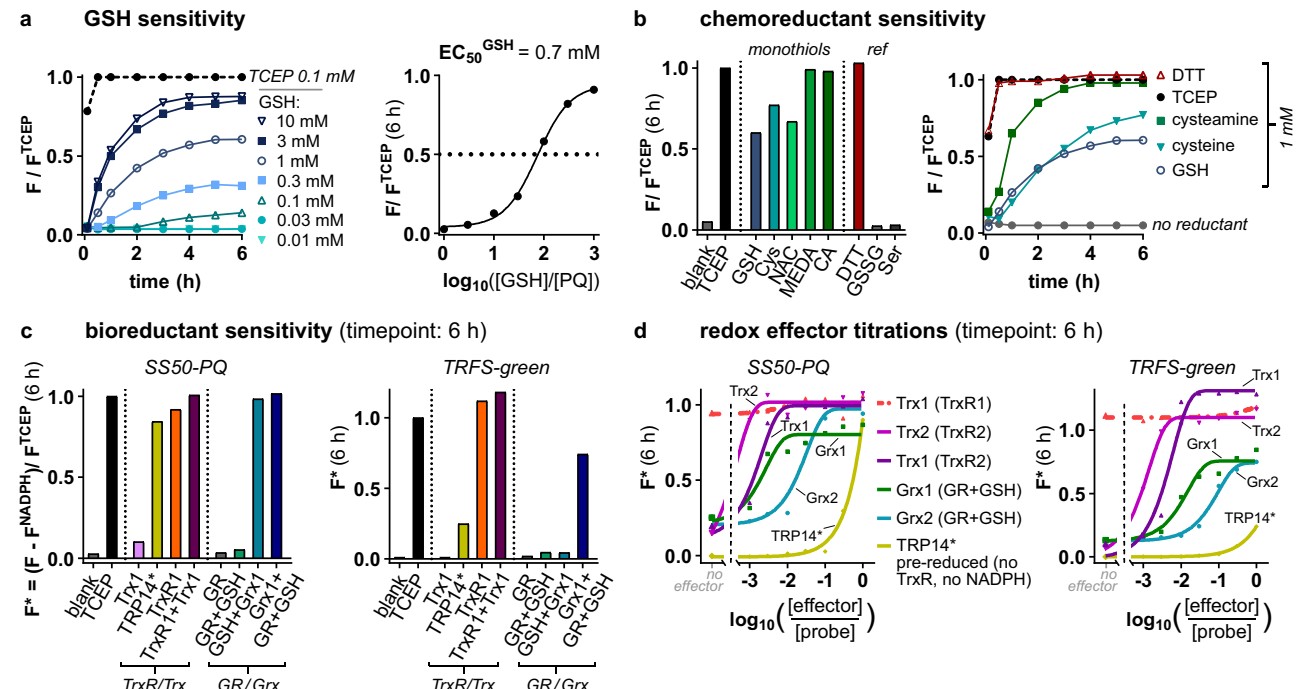

**Fig. 3 1,2-dithiolane probes are activated by a range of chemical and biological reductants. a** GSH challenge. Fluorescence timecourses of **SS50-PQ** exposed to GSH, and the corresponding dose-response plot ($t = 6$ h at 37 °C). **b** Chemical reductant assays. Normalised signal from **SS50-PQ** challenged with monothiols (GSH, cysteamine (CA), cysteine (Cys), and mercaptoethyl-dimethylamine (MEDA) each at 1 mM), dithiol (1 mM dithiothreitol (DTT)), non-reductants (GSH disulfide (GSSG) and serine (Ser) each at 1 mM), or tris(2-carboxyethyl)phosphine (TCEP) (100 μM); showing all endpoint results (6 h at 37 °C) and selected kinetics. **c** Redox enzyme assays. Normalised signal from **SS50-PQ** and **TRFS-green** challenged with TrxR/Trx or GR/GSH/Grx network proteins (20 nM TrxR1/GR, 10 μM Trx1/Grx1/TRP14*, 100 μM GSH as indicated; 100 μM NADPH in all TrxR/GR assays). **d** Dose–response plots for redox effector proteins (20 nM TrxR1/TrxR2/GR, 0.03–10 μM Trx1/Grx1/TRP14*, 100 μM GSH as indicated; 100 μM NADPH in all TrxR/GR assays). (**a–d**: data are for single representative examples from either 2 (**b**) or 3 (**a**, **c**, **d**) independent experiments; probes at 10 μM in aqueous TE-buffer; notation TRP14* indicates pre-reduced TRP14 (see "Methods").

We had hypothesised that the reducible trigger of a trigger-cargo probe is the key determinant of its reactivity and selectivity, so that results from the fast-response, environment-independent 1,2-dithiolane **SS50-PQ** should be valid for any cargo-releasing 1,2-dithiolane probes (Supplementary Note 1). To test this, we synthesised the 1,2-dithiolane **TRFS-green**[16], which has ca. 200-fold slower kinetics of releasing its aniline cargo following TCEP reduction than does **SS50-PQ** for releasing its phenol (Supplementary Fig. 5); and we challenged both **SS50-PQ** and **TRFS-green** with proteins from these redox systems (note: **TRFS-green** also suffers environment-dependency of fluorescence signal, discussed in Supplementary Note 4). To study whether probes were reduced by the effectors Trx or Grx, and/or by direct reaction with the upstream reductants TrxR or GR, we compared assays using both effectors and upstream reductants, against assays employing only upstream reductants or only effectors (TrxR/GR assays included NADPH; GR + Grx assays included 100 μM GSH for Grx reduction; see Supplementary Information).

Both 1,2-dithiolane probes were nonspecifically reduced, with Trx1, Trx2, TRP14, Grx1, Grx2 and TrxR1, all giving high rates of conversion as compared to the TCEP benchmark (Fig. 3c). Only the mitochondrial TrxR2 isoform and the GSSG-specific enzyme GR gave no signal (Supplementary Figs. 4, 5). Timecourse measurements and redox effector dose–reponse plots showed that the Trx system and the Grx-coupled GSH system (TrxR/Trx and GR/GSH/Grx; Fig. 3d, Supplementary Figs. 5–7) have identical activation profiles for both 1,2-dithiolane-based probes regardless of their cargos' leaving group character or release kinetics.

Taken together, the 1,2-dithiolane probes are nonselectively and nonenzymatically triggered by GSH and other monothiols at

subphysiological concentrations, as well as by a broad range of dithiol/disulfide-type proteins and enzymes. The systematic variation and titration of chemo- and bioreductants, and the examination of both timecourse and endpoint data, show that 1,2-dithiolanes are not TrxR-selective substrates in cell-free settings.

**General cellular and in vivo performance of the phenolate-releasing SS50-PQ probe design.** To use **SS50-PQ** to report on 1,2-dithiolane performance in general, relies on showing the technical suitability of its phenolic 2-mercapto-secondary amine carbamate design, to act as a robust reporter for redox performance of its trigger, in a variety of settings. Before tackling cellular selectivity, we therefore tested its general biological performance.

We applied **SS50-PQ** in HeLa cervical cancer, A549 lung cancer, Jurkat T-cell lymphoma, and mouse embryonic fibroblast (MEF) cell lines. All cell lines rapidly generated well-defined fluorescent precipitates of **PQ-OH**. Fluorescence platereader quantification showed a nearly linear (Fig. 4a), concentration-dependent (Fig. 4b) increase of signal, indicating that no saturation effects are operative (compare to **TRFS-green**, which saturates at 10 μM: discussed in Supplementary Fig. 8). The solid precipitates of **PQ-OH** were intracellularly localised and visible in most cells (Fig. 4c; Supplementary Movie 1).

Because microscopy can misrepresent population-level response, we used flow cytometry to collect single-cell-resolved statistics of probe activation. Though this is unusual for chemical probes, it is possible with **SS50-PQ** because the solid **PQ-OH**

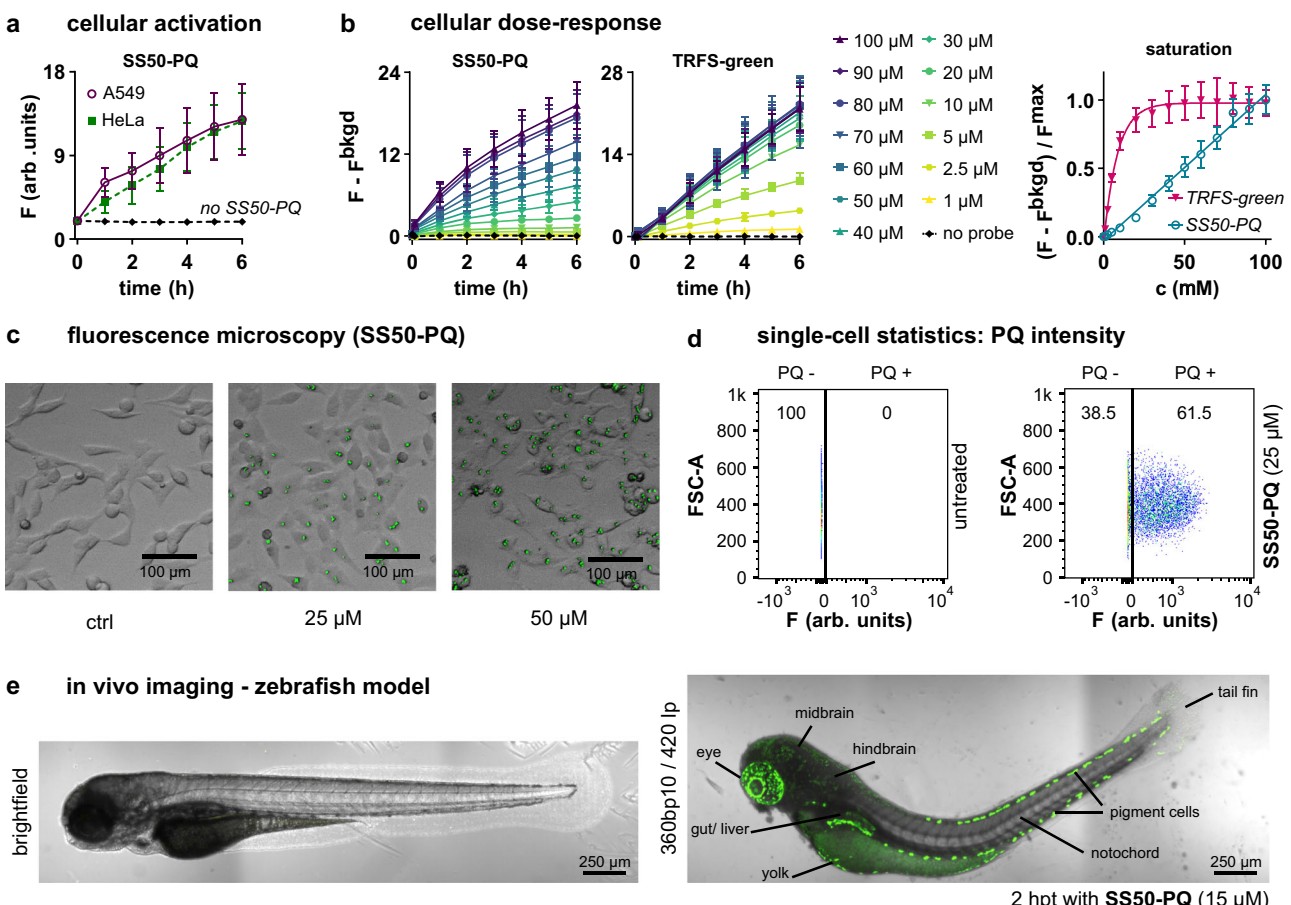

**Fig. 4 The phenolic carbamate design of SS50-PQ gives reliable performance across biological assays. a** Cellular fluorescence timecourses of **SS50-PQ** in HeLa and A549 cells (50 μM probe). **b** Dose-dependency of cellular fluorescence timecourses with **SS50-PQ**, as compared to **TRFS-green** (HeLa cells). (**a**, **b**: data as mean ± SD of ≥3 independent experiments). **c** Microscopy of **SS50-PQ**-treated HeLa cells shows fluorescent intracellular solid precipitates of released **PQ-OH** (representative images from 3 independent experiments with similar results). **d** Flow cytometry-based single-cell statistics of cellular fluorescence after **SS50-PQ** treatment (25 μM; Jurkat T-cells). (**c,d**: representative examples of ≥3 independent experiments with similar results). **e** Fluorescence imaging of embryonic zebrafish before and after **SS50-PQ** treatment (representative example of 3 independent experiments). (**c, e**: brightfield transmission image in greyscale, fluorescence superimposed in green).

precipitate is cellularly retained during fixation. These data showed a monomodal fluorescence intensity distribution with ca. >60% of cells exhibiting strong **PQ-OH** fluorescence (Fig. 4d, Supplementary Fig. 14).

We finally applied **SS50-PQ** in vivo, to stringently test three goals for its general design: (a) zero signal background, due to mechanistic quenching in the probe and to the high Stokes shift of the released fluorophore; (b) no spontaneous cargo release, due to the hydrolytic robustness of the tertiary carbamate; and (c) cellular retention of **PQ-OH** precipitates; which combine to enable high-spatial-resolution imaging in vivo. We incubated zebrafish zygotes and embryos up to 3 days post fertilisation (dpf) with **SS50-PQ** during live epifluorescence and confocal microscopy imaging (Fig. 4e, Supplementary Figs. 17, 18). Probe activation began within two hours, with interesting cell-type-specificity of the marked cells (which we do not believe is connected to differences in TrxR activity). All three probe-design goals were achieved, so that high-contrast images were obtained without background manipulation, and with precise resolution: which marks the redox probe design we report as valuable for future adaptations with other triggers.

**1,2-dithiolane probes are not cellular reporters of TrxR**. We then tested the TrxR-specificity of cellular activation of both 1,2-

dithiolane probes, using the TrxR-independent linear disulfide **SS00-PQ**[9] and the strongly TrxR-dependent selenenylsulfide-containing **RX1**[68] as references to indicate the expected outcomes of selectivity-testing cellular experiments.

Cells cultured without selenium supplementation do not fully incorporate Sec in TrxR, lowering cellular TrxR activity[69]. However, $Na_2SeO_3$ starvation or supplementation did not significantly affect **SS50-PQ** or **TRFS-green** signal (Fig. 5a, Supplementary Fig. 8a). Another method to modulate TrxR activity is to supply thiophosphate, which promotes cysteine insertion at Sec-encoding UGA codons during selenoprotein synthesis[69]. Again, we saw no effects on **SS50-PQ** or **TRFS-green** signal timecourses (Fig. 5b, Supplementary Fig. 8b). These results mirror those for the TrxR-independent probe **SS00-PQ**.

Next, we more stringently evaluated TrxR1-dependency comparing a TrxR1 knockout MEF cell line (TrxR1$^{-/-}$)[70] to its parental cell line (TrxR1$^{fl/fl}$) and using its vector-based TrxR1-knock-in line[71] TrxR1$^{2ATG}$ as an additional control. Knockout did not affect signal from **SS50-PQ** (or **SS00-PQ**), and only reduced that of **TRFS-green** by ca. 40% (Fig. 5c, Supplementary Fig. 8c), in assays that are otherwise entirely kinetically intercomparable (same cell count, parental cell line, timecourse, assay plate etc). This indicates that the dithiolane probes are indeed not effective reporters of TrxR activity in cells, as non-TrxR signal activation proceeds at highly comparable rates (see

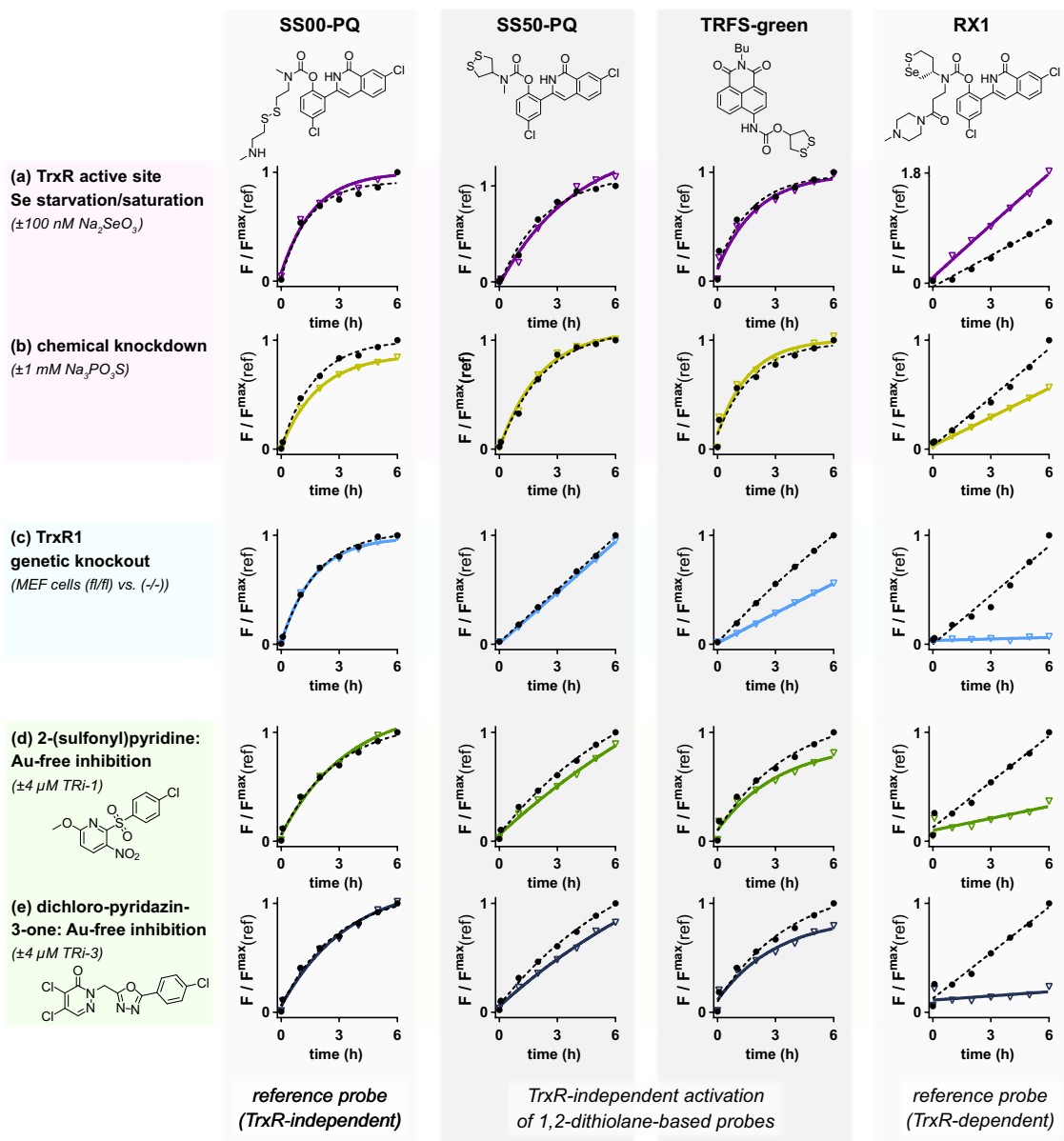

**Fig. 5 Cellular signal from 1,2-dithiolane probes does not report on TrxR. a–e** Cellular fluorescence timecourse studies of **SS50-PQ** and **TRFS-green**, with the TrxR-independent **SS00-PQ** and the TrxR-dependent **RX1** as benchmarks for assay outcomes. **a** A549 cells starved of, or supplemented with, selenium (as $Na_2SeO_3$) (probes at 100 µM; full data in Supplementary Fig. 8a). A549 cells treated with $Na_3PO_3S$ (probes at 100 µM; full data in Supplementary Fig. 8b). **c** TrxR1-knockout ($-/-$) and -wildtype (fl/fl) MEF cells (probes at 100 µM; full data in Supplementary Fig. 8c). **d**, **e** A549 cells pre-incubated for 2 h with the gold-free TrxR inhibitors TRi-1 and TRi-3 (probes at 50 µM; full data in Supplementary Fig. 10–13). (**a–e**: data shown as mean from 3 independent experiments; full representations in Supplementary Information). Note: plots on the same graphs are run under identical conditions except for the indicated variable (TrxR knockout/presence, etc.) so are kinetically comparable).

Supplementary Note 1 and discussion at Supplementary Fig. 8). The benchmark TrxR-selective **RX1** probe was instead silent in knockout cells (further evaluations in ref. [68]).

Literature claims for cellular TrxR-selectivity of probes have substantially relied on chemical inhibitor treatments that suppress their cellular activation. We first tested the recently-developed TrxR inhibitors TRi-1 and TRi-3[72]. **SS50-PQ** or **TRFS-green** had similar signal in cells pre-incubated with TRis as in untreated controls: behaviour characteristic for the TrxR-independent **SS00-PQ**, but unlike the strongly suppressed benchmark **RX1** (Fig. 5d, e, Supplementary Figs. 10–12). These results were independent of the pre-incubation time, and of whether results were acquired as population-average timecourses or as single-cell-resolved statistics (Supplementary Figs. 14, 15).

Therefore, neither TrxR suppression nor TrxR knockout greatly alter the cellular activation of **SS50-PQ** or **TRFS-green**. Taken together with the cell-free results showing rapid and nonspecific activation of the 1,2-dithiolane probe by a range of cellular thiols, we conclude that cellular activation of 1,2-dithiolane probes does not meaningfully report on TrxR activity.

**Auranofin assays can be misleadingly interpreted with 1,2-dithiolane probes.** Some previous studies of 1,2-dithiolane probes have claimed excellent cellular TrxR selectivity on the basis that treating cells with the thiophilic gold complex auranofin (AF; Supplementary Fig. 12) dose-dependently reduces fluorescence signals as compared to untreated controls[52]. AF is

popularly used as an inhibitor of TrxR; and it binds TrxR in cell-free assays as well as in cells. However, AF is a broadly "potent thiol-reactive species"[73] with at least 20 other known thiol protein targets, and is likely to bind many more depending on target exposure[74].

Notably, AF is a particularly strong binder of membrane thiols, possibly driven by its lipophilicity as well as the immediate exposure of these thiols to extracellularly-administered AF[75]. Given the extensive research showing e.g., 40-fold enhancement[30] of cellular uptake by reaction of free membrane thiols[76] with strained disulfides, we hypothesised that inhibition of dithiolane probe signal with AF treatment might simply report on uptake inhibition, rather than relating to the portion of AF which may bind to TrxR. Matching these expectations, cells pretreated with AF (0.1–4 μM) gave decreased processing of dithiolane probes but not of linear benchmark **SS00-PQ**, seen by both single-cell (Supplementary Figs. 10–13) and population average (Supplementary Figs. 14, 15) measurements.

As the dithiolanes are neither cell-free- nor cellularly-selective reporters of TrxR, this matches emerging literature to suggest that auranofin (or indeed, any other likely membrane-thiol-reactive species) in cellular studies do not test putative reductant-selectivity of 1,2-dithiolane probes (or other compounds liable for thiol-based uptake)[28]. In future work, it would be beneficial to take precautions e.g. cell-free controls testing AF-probe interactions (see Supplementary Fig. 9, with nonstrained reference **SS66C-PQ**), and comparisons to heavy-metal-free inhibitors, to check intracellular targets of strained disulfides without relying on auranofin (see Supplementary Note 2).

**The tendency of 1,2-dithiolanes to oligomerise can also be misinterpreted as probe activation.** We had seen during synthesis and handling that the 1,2-dithiolane probes and intermediates suffer from concentration-dependent, non-reductive ring-opening polymerisation (ROP), as has been extensively studied by Whitesides[24,25], and applied by Matile in SOSIPs[27]. Polydisulfides of **SS50-PQ** are fully nonfluorescent, since the ESIPT responsible for their fluorescence cannot take place without a phenolic hydrogen: thus ROP cannot trigger misinterpretable signal generation in **SS50-PQ**. As **PQ-OH** is a solid-state fluorophore, its fluorescence is also environment-independent. However, neither is true for prior art 1,2-dithiolane probes. N-acylated 8-aminonaphthilimide (**TRFS-green**) and 4-aminocoumarin (**Fast-TRFS**) both contain environment-sensitive fluorophores which are not fully fluorescence-quenched either in the N-acylated probe or in their reduction intermediates. As we had already shown the cellular non-selectivity of **TRFS-green** experimentally (discussions at Supplementary Fig. 3; and 8, and Supplementary Note 4), we now tested **Fast-TRFS** to see if ROP might operate and cause misinterpretation of its performance.

**Fast-TRFS** is a 1,2-dithiolane-based probe that does not release a cargo upon reduction (Fig. 6a)[18]. Due to partial PET-quenching from the strained 1,2-dithiolane, **Fast-TRFS** has weak fluorescence, that is partially enhanced in apolar media. Its fluorescent reduction product **dithiol-TRFS** does not have PET quenching (Fig. 6a), and its fluorescence is again much stronger in apolar media.

We asked the questions: Can non-reductive strain-promoted ROP of **Fast-TRFS** lead to signal generation under typical assay conditions? If so, can it have previously been misinterpreted as reporting on intracellular probe reduction?

Cellular assays where a relatively large surface area of lipid membrane is accessible, present a very different environment than cell-free assays in homogenous aqueous media. For hydrophobic species, and particularly for environment-sensitive fluorophores, we reasoned that this difference might be

significant. Therefore, we considered small lipid/phospholipid vesicle suspensions as a useful cell-free model to capture some aspects of inhomogenous cell-assay environments, and to test the behaviour of environment-sensitive compounds, without the complexity of cellular reductants (Fig. 6b).

Our first hypothesis was that hydrophobic **Fast-TRFS** may concentrate from aqueous media into high-surface-area apolar environments, thus initiating concentration-dependent, strain-promoted ROP of its dithiolane, giving polymer **PolyLinear-TRFS** (Fig. 6c). This is conceptually similar to Matile's SOSIPs, which exploit π-stacking of species similar to **TRFS-green** to raise local concentrations of 1,2-dithiolanes and so initiate polymerisation[27]: except that the organising principle in our hypothesis is based on partitioning out of an aqueous phase. We expected that the rate of at-membrane polymerisation of **Fast-TRFS** would depend on the available membrane surface area: i.e., the higher the vesicle concentration, the faster that ROP would proceed.

Our second hypothesis was that the fluorescence of the non-strained ROP polydisulfide product **PolyLinear-TRFS** would mimic that of non-strained monodisulfide analogues like **Linear-TRFS** (Fig. 6a): i.e., it would be more fluorescent than **Fast-TRFS** since no more PET quenching would operate. We expected that **Linear-TRFS** would also increase in fluorescence intensity with higher lipid vesicle concentration, until reaching a limit where all **Linear-TRFS** would be extracted into the lipid phase. We noted that the **PolyLinear-TRFS** would be much more hydrophobic than the morpholine **Linear-TRFS** and so might be more fluorescent in low- or zero-lipid systems, through a combination of more efficient extraction, plus the possibility of self-aggregation to exclude water and maximise fluorescence. The key prediction arising, is that the fluorescence observed during any fluorogenic ROP of **Fast-TRFS** into **PolyLinear-TRFS** should approach a similar limit as would be seen almost immediately with **Linear-TRFS**, as long as the lipid content is high enough. Relatedly, assuming **Linear-** and **Fast-TRFS** would initially partition similarly into lipid phases, we expected that in a vesicle concentration range where lipid content limits **Linear-TRFS** fluorescence, the rate of fluorescence increase of **Fast-TRFS** would also be vesicle-limited.

To test these experimentally, we synthesised **Fast-TRFS**[18] and the water-soluble analogue **Linear-TRFS** (see Supplementary Information). We prepared a vesicle suspension stock by sonicating commercial soybean lecithin (ca. 60% phospholipid, 35% lipid, with no known reductants or TrxR or NADPH) in water[77,78], then incubated stock dilutions with **Fast-TRFS** and **Linear-TRFS**.

In brief, these incubations showed strong vesicle-dependent fluorogenicity of **Fast-TRFS** in the absence of any reductant, which within an hour reached the same fluorescence values as quantitative reduction with TCEP. The data were consistent to all predictions arising from the hypotheses of non-reductive signal generation by partitioning-and-ROP-based fluorogenicity. This argues that cellular assays of environment-dependent 1,2-dithiolane probes may have suffered particular problems of misinterpretation:

At <1%wt vesicles, the lipid content controlled **Linear-TRFS** fluorescence, which plateaued after <5 min regardless of vesicle concentration, coherent with a fast extraction equilibrium into a fluorescence-enhancing apolar environment (Fig. 6d). In the first 5 min, a fast increase in **Fast-TRFS** fluorescence also reached a lipid-content-dependent value, also coherent with fast extraction of the intact dithiolane to initial equilibrium in lipid, though with its environment-dependent fluorescence increase being smaller. Then, however, fluorescence of **Fast-TRFS** continued to increase, at slower rates that were also dependent on lipid content:

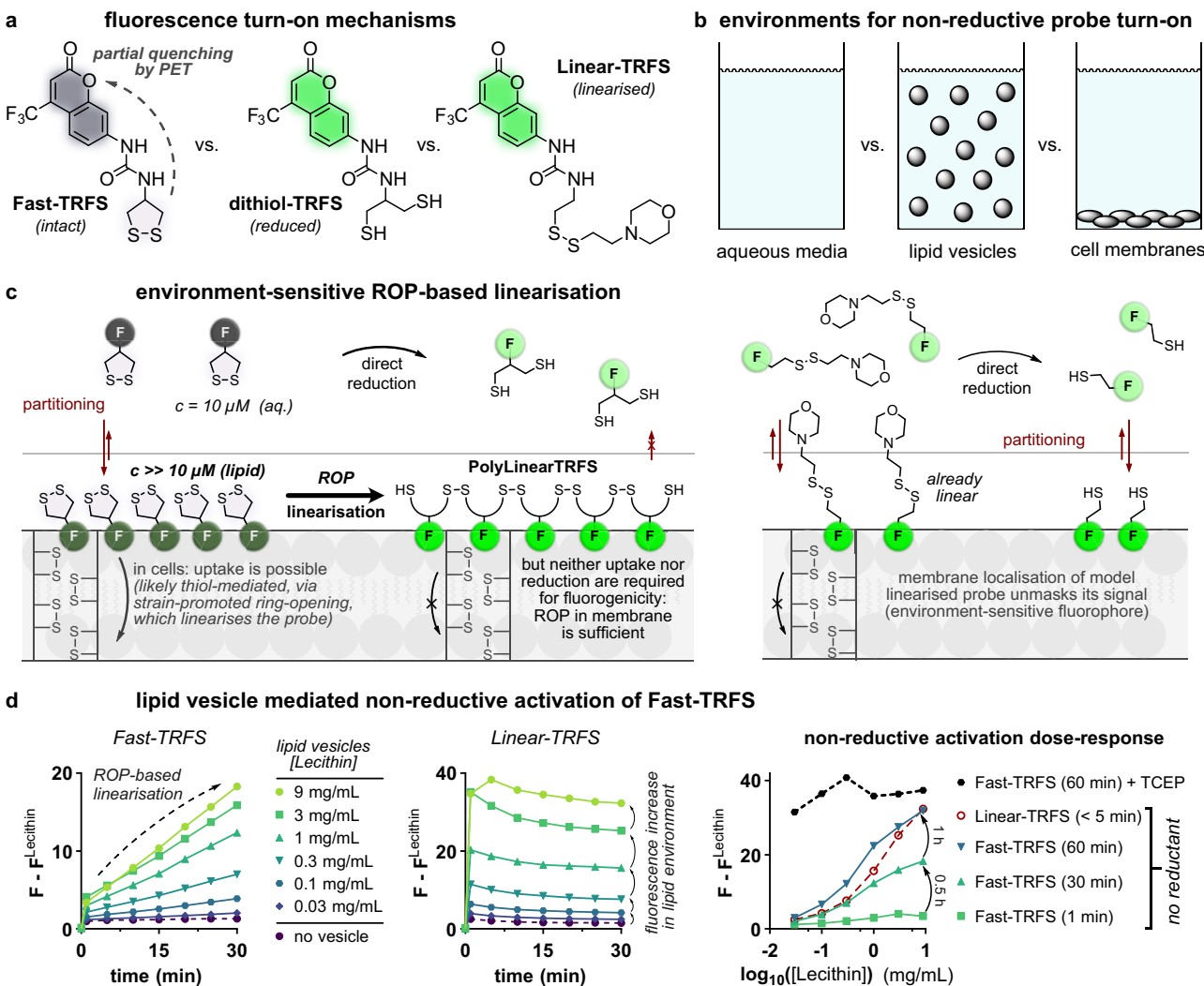

**Fig. 6 The fluorescence of non-cargo-releasing 1,2-dithiolane Fast-TRFS activates in the absence of TrxR and of reductants, after partitioning into membranes: coherent with strain-promoted oligomerisation and environment-dependent fluorescence. a** Weak, environment-dependently fluorescent **Fast-TRFS**[18]; and strong environment-dependently fluorescent compounds **dithiol-TRFS** (its reduction product) and **Linear-TRFS** (a reference compound). **b** The inhomogenous media of cellular assays may be closer modelled with lipid vesicle suspensions than as an all-aqueous system. **c** Extractive concentration of **Fast-TRFS** into lipid membranes may drive concentration-dependent, strain-promoted ring-opening polymerisation (ROP) to nonstrained polydisulfide **PolyLinear-TRFS**, whose fluorescence may mimic that of nonstrained monodisulfide **Linear-TRFS**. Thus, ROP may be a non-reductive mechanism for fluorescence turn-on of **Fast-TRFS**. **d** Fast-TRFS and **Linear-TRFS** (each 10 μM) incubated with lecithin vesicles. After 60 min, TCEP (0.1 mM) was added to benchmark complete reduction. (Representative example from 2 independent experiments; full data in Supplementary Fig. 16).

coherent with slower fluorogenic oligomerisation of **Fast-TRFS**, for which the available lipid volume plays the role of catalyst loading. In just one hour, the **Fast-TRFS** signal in the presence of 0.9%wt vesicles but without reductants, reached the same fluorescence plateau as established by adding 10 eq. of TCEP, which pleasingly was the same signal as seen within ≤5 min of mixing **Linear-TRFS** with 0.9%wt vesicles (Fig. 6d).

These data are fully coherent with our hypotheses of **Fast-TRFS** undergoing non-reductive ROP to **PolyLinear-TRFS** in inhomogenous media, that can easily reach maximum theoretical fluorescence values. We also noted that addition of TCEP to **Linear-TRFS** (giving the monothiol) did not change its fluorescence (Supplementary Fig. 16b). Thus, in the inhomogenous-media settings of cellular or lysate assays, **Fast-TRFS** likely cannot distinguish reduction to **Dithiol-TRFS**, from thiol-disulfide exchange to a monothiol product, or from non-reductive ROP to **PolyLinear-TRFS**. **Fast-TRFS** signal is likely to be dominated just by the environment-dependency of

fluorescence that each of these three species displays. While these cell-free data do not test the fate of **Fast-TRFS** in the cellular setting, we feel that it is likely that confounding factors operating in simple models will be even more problematic in complex ones (see Supplementary Note 3).

In summary, we believe that this hitherto-unreported reduction-independent fluorogenicity of the 1,2-dithiolane probe, combined with the orthogonal demonstrations of cell-free and cellular nonspecificity shown for the robustly-interpretable **SS50-PQ** as well as for **TRFS-green**, argue convincingly that 1,2-dithiolane-based probes should not be interpreted as being selective reporters for any specific reductase in lysates, cells or in vivo.

## Discussion

Dithiol/disulfide-exchange reactions are central to biology, and engineered disulfides exploiting these reaction manifolds are

finding applications from chemical biology probes to biophysics and materials chemistry. Linear disulfides have been known for decades as substrates for nonspecific thiol attack/reduction, and often used for intracellular release of cargos. 1,2-dithiolanes have emerged as substrates of interest in chemical biology[16–18], although it has remained contentious whether their reduction is nonspecific, or is selectively performed by TrxR. We answered this question by studying the biochemical and biological performance of the 1,2-dithiolane-based redox probe **SS50-PQ**, where the disulfide was integrated in a stable, modular design, with a release-activated fluorogenic cargo, that we could even use in flow cytometry studies and for cell-resolved imaging in live embryos. We used nonselective linear disulfide **SS00-PQ** and TrxR-selective probe **RX1** as negative and positive benchmarks in our assays; and to show the general applicability of the results, we also performed comparisons to the known 1,2-dithiolane probes **TRFS-green** and **Fast-TRFS**, introducing further compounds (**Linear-TRFS** and **SS66C-PQ**) as needed to study aspects of the fluorophores.

We have demonstrated the nonspecific cleavage of 1,2-dithiolane probes through a rigorous methodology of reductant titrations; cell-free enzyme screenings; and cellular knockout/knockdown/knock-in, activity suppression, and electrophilic inhibitor experiments, which we believe will be a valuable methods addition to the literature. While the probes can indeed be rapidly opened by TrxR[19], they are opened just as rapidly by nearly every other reducing thiol species we tested (Fig. 3, Supplementary Fig. 7), most of which are expressed at much higher concentrations in cells. We conclude that in cells, they do not selectively report on TrxR. We also conclude that this is an intrinsic feature of the 1,2-dithiolane motif itself, independent of the cargo to which it is attached and of the chemical manner of the attachment (Fig. 5). This is coherent with recent reports stating that **TRFS-green** and **Fast-TRFS** are cellularly reducible by Trx and Grx[53], although our assays support a much stronger statement: that nonspecific, and very likely cell-surface, reduction by thiols - and not reaction with any limited set of intracellular reductases - is likely responsible for cellular signals from 1,2-dithiolane probes.

We have revealed features of previous probes and experimental designs which may have been misleadingly interpreted in the field as showing TrxR-selectivity. These include: (1) environment-dependent fluorogenicity of intact probes (Supplementary Note 4); (2) concentration-dependent, strain-promoted, non-reductive oligomerisation of 1,2-dithiolane probes which in the case of **Fast-TRFS** fully unquenches its fluorescent signal upon concentration into lipid membrane environments (Fig. 6 and Supplementary Note 3); (3) thiol-coordinating or thiol-alkylating reagents such as auranofin may suppress cellular signal generation by blocking thiol-mediated, strain-promoted cellular uptake.

Taken together, we conclude that 1,2-dithiolane is an easily and nonspecifically thiol-opened/-reduced motif which is not a TrxR-selective substrate in cells; and that 1,2-dithiolane-based probes are not selective TrxR reporters in cellular settings.

We note however, that there may be immediate rewards if the growing literature of 1,2-dithiolane-based probes is re-evaluated[53]. This would (i) maintain a clear literature; (ii) avoid that nonspecific electrophilic pan-assay interference compounds which suppress dithiolane probe signal by blocking thiol-mediated uptake are falsely identified as TrxR-substrate hits (PAINS: see Supplementary Note 1); and could perhaps (iii) allow 1,2-dithiolane-derived prodrugs instead to find uses as modular systems for thiol-mediated cellular uptake and activation, impacting research in cell penetration and assisted uptake (discussion in Supplementary Note 2).

Diversifying trigger structures to reach redox substrates that *are* selective for key oxidoreductases remains a central goal for research in the field. By identifying and avoiding problematic and nonselective substrate types such as 1,2-dithiolane, chemical development may instead be oriented towards selective and robust redox chemotypes for bioreductive probe and prodrug research. Indeed, novel reducible motifs with chemotype-based selectivity, as in the recent Trx-selective **SS60-PQ**[9] and the TrxR-reporting probe **RX1**[68], are just now emerging. The modular reduction-triggered phenolic carbamate system we developed for this work already ensures valuable performance in biology, through its zero signal background, excellent hydrolytic robustness, and retained cell-marking signal, which combine to offer high-spatial-resolution imaging and cell-resolved statistics (Fig. 4). As selective reduction-sensing units for e.g. for GR, TrxR or Trx are identified, installing them as trigger motifs in this modular system would retain these beneficial features: giving powerful and useful probes for redox biology that can at last allow researchers to unveil the dynamics of these major dithiol/disulfide-type enzyme systems within cells. Such probes are already being reported and further developments will be communicated in due course.

## Methods

For detailed experimental protocols and further information on all 'Methods', see Supplementary Information.

**Probe synthesis and characterisation**. Reactions and characterisations were performed by default with non-degassed solvents, reagents and building blocks, that were purchased from standard commercial sources. Anhydrous solvents obtained in septum-capped bottles and analytical grade or higher quality solvents were used without purification. Industrial grade solvents were distilled prior to use. Unless otherwise stated, reactions were performed at room temperature without precautions regarding air or moisture and were stirred using magnetic Teflon®-coated stir bars. Air or moisture sensitive reactions were conducted in dry Schlenk glassware. Flash column chromatography was performed on Geduran® Si 60 silica gel from Merck GmbH, Darmstadt (Germany) or using a Biotage® Select automated column chromatography system with manufacturer's cartridges from Biotage GmbH, Uppsala (Sweden). Thin layer chromatography to monitor reactions and determine $R_f$-values was performed on silica coated aluminium sheets with fluorescent indicator (TLC Si 60 F254 from Merck GmbH, Darmstadt, Germany) with visualisation by UV irradiation (254 nm/360 nm) or staining with $KMnO_4$ solution (3.0 g $KMnO_4$, 20 g $K_2CO_3$, 0.30 g KOH, 0.30 L $H_2O$).

**General fluorescence assay methods**. Fluorescence readout of cell-free activity and/or cell assays was performed either using a FluoStar Omega plate reader from BMG Labtech, Ortenburg (Germany) (ex/em 355bp10/520lp or 440bp10/520lp) using Omega Software version 5.50 R3 from BMG Labtech, Ortenburg (Germany) or a Tecan Infinite M200 plate reader with integrated software from Tecan, Maennedorf (Switzerland) (ex/em 355bp10/520lp or 440bp10/520lp) recording fluorescence intensity. Reduction-mediated activation of **SS50-PQ** releases the water-insoluble dye **PQ-OH**$_{(s)}$ with precipitation-based induction of ESIPT-based fluorescence (typically monitored with ex/em 355bp10/520lp). Reduction-mediated activation of **TRFS-green** releases 1-butyl-6-amino-naphthalimide (typically monitored with ex/em 445bp10/520lp). For **TRFS-green**, subtraction of background fluorescence was conducted as indicated in the Supplementary Information. We noted that upon standing, dithiolane probe stocks showed decreasing capacity to generate fluorescence in reductant assays. Consequently, stocks were prepared freshly for all assays. All reactions were incubated at 37 °C and 100% humidity. Timecourse measurements were conducted to determine kinetics of reduction-mediated release. Data were typically interpreted by normalising the absolute, time-dependent fluorescence intensity F(t) to the peak theoretical signal observed in a maximum-fluorescence experiment using 100 μM of TCEP ($F^{TCEP}(t)$) to remove the influence of cyclisation/release/precipitation kinetics from analysis. For cell-free bioreductant assays, F(t) was pre-corrected by subtracting the absolute time-dependent background fluorescence $F^{NADPH}(t)$ caused by auto-fluorescence of reduced $\beta$-NADPH in the assay. Dose–response curves were fitted using GraphPad Prism version 8.0.2. from GraphPad Software, San Diego (USA).

**Cell-free chemoreductant assays**. In a black 96-well plate with black bottom, 80 μL of a diluted solution (12.5 μM in aq. TE, pH 7.4, 1.25% DMSO) of **SS50-PQ** (final concentration 10 μM) was mixed with a solution of selected chemical reductants (20 μL of a solution (50 μM to 50 mM in aq. TE, pH 7.4). Reductants used include tris(carboxyethyl)phosphine (TCEP) at 100 μM final concentration, and dithiothreitol (reduced, DTT), glutathione (reduced, GSH), glutathione disulfide (oxidized, GSSG), cysteine (Cys), serine (Ser), mercaptoethyl-dimethylamine

(MEDA), cysteamine (CA) and *N*-acetylcysteine (NAC) all at 1 mM final concentration for single-concentration profiling; or e.g. 10 µM, 30 µM, 100 µM, 300 µM, 1 mM, 3 mM and 10 mM reductant for dose-response characterisation.

**Cell-free inhibition experiments**. 80 µL of a diluted solution (62.5 µM in aq. TE, pH 7.4, 1.25% DMSO) of **SS50-PQ** (final concentration 25 µM or 50 µM as indicated) was pre-treated with 2-((4-chlorophenyl)sulfonyl)-6-methoxy-3-nitropyridine (TRi-1), 4,5-dichloro-2-((5-(4-chlorophenyl)-1,3,4-oxadiazol-2-yl)methyl)pyridazin-3(2*H*)-one (TRi-3)[79] (from DMSO stock solutions) or auranofin (AF) (from ethanol stock solutions) with final inhibitor concentrations from 0.1 µM to 4.5 µM.

**Cell-free bioreductant assays**. In a black 96-well plate with black bottom, 50 µL of a diluted solution (20 µM in aq. TE, pH 7.4, 2% DMSO) of **SS50-PQ** or **TRFS-green** (final concentration 10 µM) was mixed with 40 µL of the selected oxidoreductase (TrxR1, TrxR2, GR to reach final concentrations of 20 nM) and/or its native substrate (Trx1, Trx2, TRP14, Grx1, Grx2 to reach final concentrations of 10 nM to 10 µM respectively), including 100 µM final concentration of GSH for the GR-GSH-Grx system assays. The reaction was started by addition of 10 µL of a solution of β-NADPH (1 mM in aq. TE, pH 7.4, reaching 100 µM final concentration). Human recombinant thioredoxin (Trx1 and Trx2) (lyophilized), human recombinant glutaredoxin (Grx1 and Grx2) (lyophilized from 10 µL TE-buffer, pH 7.5), human recombinant thioredoxin-related protein of 14 kDa (TRP14), human thioredoxin reductases (TrxR1 and TrxR2) (1.5 m/mL in 50% glycerol/TE buffer, pH 7.5) and human recombinant glutathione reductase (GR) (100 µM in 50% glycerol/TE buffer, pH 7.5) were obtained from IMCO Corp., Stockholm (Sweden) or produced and purified as previously described[80,81].

**Cell-free environment-dependent polymerisation assay**. Commercial α-lecithin (CAS 8002-43-5, Sigma-Aldrich P5638) was hydrated with ultrapure water (Milli-Q, Reptile Bioscience Ltd., Boston, MA) to a final concentration of 9 mg/mL. The suspension was gently vortexed to achieve a homogeneous phase, which was then subjected to one freeze/thaw cycle. The stock solution was extruded 21 times through a polycarbonate membrane with a pore diameter of 400 nm, using a Mini Extruder (Avanti Polar Lipids, Inc., Alabama, United States) according to Fromherz et al.[77]. The vesicle stock was then diluted to the indicated concentrations and treated with DMSO stock solutions of **Fast-TRFS** and **Linear-TRFS** to final probe concentrations of 10 µM or 100 µM with maximum 1% DMSO. 1-octadecanethiol (RSH) was optionally spiked at catalytic (0.05 eq) or excess (1.5 eq vs. probe concentration) amounts. Fluorescence intensity (355bp10/460bp20) was recorded over time and compared to an untreated reference experiment (vesicle scattering background). The potential for further fluorescence increase mediated by reduction of intact monomolecular dithiolane was tested by subsequent treatment of the system with excess TCEP (0.1 mM, i.e. 1-10 eq.).

**General cell culture**. Cells were grown at 37 °C under 5% CO₂ atmosphere and cell growth was confirmed using a Nikon Eclipse Ti microscope (Nikon Corp., Minato (Japan)). HeLa (DSMZ; ACC57), A549 (DSMZ; ACC107) and Jurkat (ATCC; TIB-152) cell lines were purchased from the German Collection of Microorganisms and Cell Cultures. Parent ("fl/fl"), TrxR1 knockout ("-/-", ko) and Sec-TrxR1 reconstituted ("2ATG") mouse embryonic fibroblasts (MEF) were a kind gift from Marcus Conrad (**ko**: MEFs isolated from conditional TrxR1 knockout mouse embryos were immortalised by lentiviral transduction. In vitro deletion of TrxR1 was achieved by Tat-Cre induced recombination and verified by PCR and Immunoblotting for TrxR1[70]. **2ATG**: lentiviral transgenic stable expression of Txnrd[498Sec] on the ko line[71]). All cell lines are tested regularly for mycoplasma contamination and only mycoplasma negative cells are used in assays. HeLa, A549 or MEF cells were grown in Dulbecco's modified Eagle's medium (DMEM: glucose (4.5 g/L), glutamine, pyruvate, phenol red, NaHCO₃ (2.7 g/L); PAN Biotech, Aidenbach (Germany)). Jurkat cells were grown in RPMI-1640 medium (glutamine, sodium bicarbonate; Merck KGaA, Darmstadt (Germany)). Media were supplemented with 10% heat-inactivated fetal bovine serum, penicillin (100 U/mL), streptomycin (100 µg/mL) and optionally with Na₂SeO₃ (0–100 nM) or Na₃PO₃S (0–1.2 mM). PBS Dulbecco buffer (Merck GmbH, Darmstadt (Germany)) was used for washing and resuspending steps; TrypLETM Express (gibco Life Technologies Inc., Massachusetts (USA)) was used for trypsination.

**Cellular activation/inhibition assays**. Cells were seeded in 96-well plates (F-Bottom, black, Fluotrack, high binding; Greiner bio-one GmbH, Kremsmünster (Austria)) in 100 µL medium. Medium was treated with **SS50-PQ**, **TRFS-green**, **SS00-PQ**[9] or **RX1**[68](from stock solutions in 100% DMSO) to reach final concentrations of 1 µM to 100 µM at maximum final levels of 1% DMSO. For inhibition experiments, cells were pre-treated with TrxR inhibitors TRi-1 and TRi-3[79] (from DMSO stock solutions) and AF (from ethanolic stock solution), 2 h, 3 h or 15 h before probe treatment. Treated cell plates were incubated at 37 °C under 5% CO₂ atmosphere. Kinetics of cellular processing were determined by timecourse fluorescence measurements, performed as in the cell-free assays, with optional background-corrected fluorescence intensity F(t)-F^Background(t) where indicated, as needed.

**Fluorescence microscopy**. Intracellular **PQ-OH**(s) precipitation was confirmed using a Nikon Eclipse Ti2 upright microscope from Nikon Instruments Europe BV, Amsterdam (Netherlands) (ex/em 355bp50/410lp; or transmitted light, as appropriate). Images were processed using Fiji version 1.51 (ImageJ). Confocal time lapse microscopy was performed on live HeLa cells seeded in 8-well slides (ibiTreat µ ibidi slides, ibidi GmbH, Martinsried (Germany)). Slides were placed on the motorized stage of a Leica SP8 laser-scanning confocal microscope (Wetzlar, Germany), treated with **SS50-PQ** at 50 µM on the stage and immediately imaged for one hour with ex/em 405laser/530bp20, collecting fluorescence and brightfield images.

**Flow cytometry-based single-cell statistics**. After treatment, cells were harvested and stained with a fixable viability dye according to the manufacturer's recommendations (zombie NIR™ Fixable Viability Kit, BioLegend). Cells were fixed in 4% paraformaldehyde (PFA) in PBS for 30 min and either stored in PBS or immediately resuspended in a wash buffer containing PBS with 1% bovine serum albumin (BSA) and 1 mM EDTA. Flow cytometry was conducted at the BioMedical Centre Core Facility of the LMU Munich on a BD LSRFortessa (BD Bioscience, Heidelberg (Germany)) using the integrated BD FACS Diva software v.8.0.1. The following excitation/detection settings were used: zombie (ex/em 647laser/780bp60) and PQ fluorescence (ex/em 355laser/530bp30). Data were processed using FlowJo v.10.7.1 (BD Biosciences). An unstained sample was used to exclude cell debris and doublets. Zombie dye was used to exclude dead cells. PQ-positive gate was set so that 0% of cells were PQ-positive in an unstained sample. Cell debris, singlets, Zombie and PQ gates were set on an appropriate sample in each experiment and applied to all samples. Gating strategy is shown in Supplementary Fig. 6.

**In vivo activation experiments**. *Danio rerio* embryos (8 hpf or 3 dpf, sex undetermined) were housed in groups of 20-40 individuals in a fish facility (Aquaneering) maintaining approx. 700 mS, pH 6.9–7.1 and 28 °C with a 14/10 h light/dark cycle as outlined by common zebrafish handling guidelines. All experiments used fertilized eggs from *ab* wild-type parents, grown in 30% Danieau medium (0.12 mM MgSO₄, 0.21 mM KCl, 0.18 mM Ca(NO₃)₂, 17.4 mM NaCl, 1.5 mM HEPES, pH 7.2) at 28 °C with a 14/10 h light/dark cycle. Where reported, to suppress melanin pigmentation, 150 µM phenylthiourea (PTU) was added to the 30% Danieau medium at 8-10 hpf. All procedures involving animals were carried out according to EU guidelines and German legislation (EU Directive 2010_63, licence number AZ 325.1.53/56.1-TU-BS). As all zebrafish embryos in this study were analysed at ages below 5 dpf, no approval by an ethics board was required. The study adhered to ARRIVE guidelines. **SS50-PQ** stocks (10 mM in DMSO) were warmed to 37 °C before use. 5 µM, 15 µM and 45 µM dilutions were freshly prepared in 30% Danieau supplemented with DMSO to 1% (final concentration). At 8 hpf or 3 dpf, embryos were transferred to each well of a 6 well plate, and the 30% Danieau medium exchanged for the test dilutions (5 mL per well). Embryos were incubated at 28 °C until the formation of solid green-fluorescent particles was observed. Signal development was initially monitored on a stereofluorescence microscope (Leica M205FA or MDG41, from Leica Microsystems, Wetzlar, Germany) equipped with a UV filter (ex/em 360bp40/420lp) and acquired using Las X software v. 5.0.3. (Leica Microsystems, Wetzlar (Germany)). Then, for confocal imaging, the embryos were embedded in lateral position in 1.2% ultra-low gelling agarose (type IX-A), overlaid with the test solution. The embryos were transferred to a Zeiss Airyscan confocal microscope (Carl Zeiss, Jena (Germany)), with a Life Imaging Services heating chamber set to 28 °C. Whole embryos were recorded using a ×10 objective and the 405 nm diode for excitation, with image acquisition controlled by Zen Black software v.2.5. (Carl Zeiss, Jena (Germany)), and subsequent tile/grid stitching in Fiji version 1.51 (ImageJ).

**Statistics**. Sample size and a description of the statistical parameters including central tendency, variation or estimates of uncertainty is provided in Figure legends. Representative experiments are shown where appropriate and indicated accordingly.

**Reporting summary**. Further information on research design is available in the Nature Research Reporting Summary linked to this article.

## Data availability

All data generated or analysed during this study are included in this article and its Supplementary Information files, including the Source Data file (raw data for Figs. 3–6 and Supplementary Figs. 3–16). These and all data of this study can also be obtained from the authors upon request. None of these datasets are resources of public interest and therefore are not archived publicly in other forms. Source data are provided with this paper.

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

## Acknowledgements

This research was supported by funds from the German Research Foundation (DFG: SFB 1032 project B09 number 201269156, SFB TRR 152 project P24 number 239283807, SPP 1926 project number 426018126, and Emmy Noether grant 400324123 to O.T.-S.; SPP 1710 project BE 1540/23-2 to K.B. (now transferred to Stefan Rahlfs)); LMUExcellent (Junior Researcher Fund to O.T.-S.); the Munich Centre for NanoScience initiative (CeNS, to O.T.-S.); and from Karolinska Institutet, The Knut and Alice Wallenberg Foundations, The Swedish Cancer Society, The Swedish Research Council and the Hungarian Thematic Excellence Programme (TKP2020-NKA-26) to E.S.J.A. J.G.F. thanks the Studienstiftung des deutschen Volkes for support through a PhD scholarship; L.Z. thanks the Fonds der Chemischen Industrie for support through a PhD scholarship; L.P. thanks the GRK 2338 for support through a PhD scholarship; J.T.-S. thanks the Joachim Herz Foundation for fellowship support. We thank Martina Ober (LMU) for extensive help preparing vesicles; Marcus Conrad (Helmholtz Centre, Munich) for MEF and knockout MEF cell lines; Hartmann Harz and Christoph Jung (LMU microscopy platforms) for assistance with microscopy facilities; and Matt Fuchter (ICL), Kate Carroll (Scripps), Rob Hondal (UVM), Stefan Matile (Uni Geneva), and the attendees of the SPP 1710 Thiol-Based Redox Switches conferences in 2019 and 2021 for their supportive and collegial discussions.

## Author contributions

J.G.F. performed synthesis, chemical analysis, chemoreductant and enzymatic cell-free studies, cellular studies, and coordinated data assembly. L.P. and K.C.S. performed enzymatic specificity screenings, cellular inhibitor studies and FACS-based single-cell statistics. S.B. and C.B. performed enzymatic specificity screenings. U.T. performed zebrafish embyro studies. L.Z. and M.S.M. performed synthesis and analysis. K.B. and E.S.J.A. supervised enzymatic specificity screenings. J.T.-S. performed cellular studies, supervised cell biology and coordinated data assembly. O.T.-S. designed the concept and experiments, supervised all other experiments, coordinated data assembly and wrote the manuscript, with input from all co-authors.

## Funding

## Competing interests

J.G.F., L.Z., and O.T.-S. are inventors on patent application EP21167187.0 filed by the LMU Munich in 2021 covering the structure of compound **RX1** which is used as a control in this paper. All authors declare no other competing interests.
