## [Peer Review File · Nature Communications]

Cyclic 5-membered disulfides are not selective substrates of thioredoxin reductase, but are opened nonspecificallyREVIEWER COMMENTS

Reviewer #1 (Remarks to the Author):

This manuscript offers some compelling evidence that several probes containing the cyclic disulfide moiety are, contrary to several prior reports, are non-selective for TrxR. Since this work is refuting prior peer-reviewed studies it is imperative that the authors carefully check that they have properly represented the challenged studies. For example:

1. The authors did not use the same probe as in the prior work. This assumes that only the disulfide plays a significant role in the selectivity, as opposed to binding in the active site. Many covalent drugs are now known that have highly reactive functional groups but that are nonetheless still efficacious for example. A more convincing case is needed to show that all cyclic dithiane probes are nonselective.
2. The AF inhibition data is important. It is reasonable to assume that the probe interaction with AF would decrease its signal. But it's not clear why the authors (apparently?) just hypothesized this rather than doing such a seemingly simple control experiment. Also, was AF inhibition the only control used in the prior work under challenge? Didn't they use any control disulfide-reacting species such as cysteine or glutathione as the current authors herein use? What were the prior results, how do they compare to the controls described herein?
3. There are other published TrxR probes that have other seemingly nonspecific functional groups such as linear disulfides, including relatively simple commercial kits (Ellman's reagent?). Why would these other reagents not be commented upon by the authors herein?

Reviewer #2 (Remarks to the Author):

The goal of this work was to verify whether the 1,2-dithiolane-based probes, claimed to be specific inhibitors of mammalian thioredoxin reductase, are indeed specific. To this end, the authors have designed a new probe based on previous work. In vitro assays clearly show that monothiols such as cysteine (Cys), N-acetylcysteine (NAC), N,N-dimethyl-cysteamine (MEDA), and cysteamine (CA) and the reductive enzymatic systems – Trx/TrxR and Grx/GSH/GR are excellent reductants of the 1,2-dithiolane moiety and generate rapid fluorescence response. In other words, the probe is not specific.

However, the fluorescence microscopy (Figure 4c), flow-cytometry (Figure 4d), and the fluorescence imaging of zebrafish (Figure 4e) don't add much to the story and may even be a distraction to some readers. Additionally, the animal study is complicated because of the expression pattern of TrxR1 at various life stages of the animal. Moreover, the determination of the mechanism by which AF inhibits the interaction between TrxR1 and the probe is critical and will add great value to this work. Whether or not AF's binding to membrane thiols actually affects the cellular entrance of the probe can be investigated by designing a similar probe in which the fluorophore is always turned on and remains attached to the 1,2-dithiolane moiety irrespective of its redox status. Also, the author should consider the possibility that when the probe enters the cell and gets reduced to the thiol form, it can react in the reduced form with AF, which, as the author mentioned, is a "potent thiol reactive species". Expectedly, the progress of the reaction can be monitored using mass spectrometry or some other method.

Response to Reviews

Reviewer 1: This manuscript offers some compelling evidence that several probes containing the cyclic disulfide moiety are, contrary to several prior reports, are non-selective for TrxR. Since this work is refuting prior peer-reviewed studies it is imperative that the authors carefully check that they have properly represented the challenged studies. For example:

1. The authors did not use the same probe as in the prior work. This assumes that only the disulfide plays a significant role in the selectivity, as opposed to binding in the active site. Many covalent drugs are now known that have highly reactive functional groups but that are nonetheless still efficacious for example. A more convincing case is needed to show that all cyclic dithiane probes are nonselective.

We address this in two major ways.

Firstly, we now supply a large catalogue of cell-free and cellular experiments comparing our SS50-PQ with the previously-published 1,2-dithiolane **cargo-releasing** probe TRFS-green (Fang et al. **2014** *J. Am. Chem. Soc.*, 10.1021/ja408792k).

(1) For the irreversible cyclisation-driven-release probe TRFS-green, it is significant to show which reductants are capable of triggering fluorophore release in the cell-free setting. As we show in the newly expanded **Fig. 3c-d**, the aniline-eliminating probe TRFS-green is kinetically labile to all tested redox effector proteins Trx1, Trx2, Grx1, Grx2 and TRP14, with very similar dose-response as the phenol-eliminating SS50-PQ, just with an apparent shift to higher reductant doses: although this shift arises not because of any difference of intrinsic reducibility (selectivity) of their 1,2-dithiolanes, but simply because anilines are poorer leaving groups. Therefore, we consider that the post-reduction-expulsion kinetics of SS50-PQ and TRFS-green are predictably different according to the nature of the chemical cargo (TCEP challenge half-life ca. 3 hours for aniline expulsion of TRFS-green, but only ca. 5 minutes for phenolate expulsion from the more sensitive and easier to interpret SS50-PQ). This similarity is supported by several pages of extra figures and data in the supporting information (new **Fig S5, S6**). We also plot full dose-responses and explicitly tabulate EC_{50} values of SS50-PQ compared to TRFS-green for all protein effectors (new **Fig S7**) to show how there is a consistent 3-to-5-fold difference of the 6 hour fluorescence dose-response across all effectors. To us this is very compelling reiteration that redox structure-activity relationships do matter: the 1,2-dithiolane is the reducible motif constant across both probes, the distant aniline/phenol cargo is the leaving group kept constant between probe series, the only way that reduction rates between the series can maintain the same ratios across different reductants is if the dithiolane motif itself dictates reactivity (i.e. to answer the reviewer: yes - the disulfide plays the key role in selectivity, not the cargo).

(2) We also take TRFS-green through cellular signal generation studies to show that its signal generation is independent from its claimed reductase TrxR, supporting our general conclusion that 1,2-dithiolane probes are not capable of cellular selectivity:

(2a) TrxR1 knockdown does not prevent TRFS-green signal generation in cells (new **Fig 5, Fig S8**) but only reduces it by ca. 40%. TrxR2, the other isozyme of TrxR, is only present in mitochondria, and is not expected to have a comparable expression level to TrxR1. Additionally, TrxR2 is also known to have lower turnover of small molecule substrates in general, and our cell-free data show (new **Fig S5**) that it is almost entirely incapable of TRFS-green reduction. Therefore at least ca. 60% of signal generation in cells by TRFS-green cannot be directly related to TrxR activity. Regarding the 40% drop, as **we note, this** can arise straightforwardly in probes that are partially processed by downstream reductants dependent on TrxR, such as Trxs, without this change indicating any direct reduction of the probe specifically by TrxR. By comparison, SS50-PQ is entirely unresponsive to TrxR1 knockout. The similarity of cellular behaviour of the two probe systems, which in cell-free settings are strikingly similarly reducible by the same broad range of redox effectors, should be kept in mind: it strongly suggests that results from SS50-PQ should also apply to TRFS-green. We believe this indicates that the SS50-PQ environment-independent release-based probe has more straightforwardly revealed, that 1,2-dithiolane probe activation is not significantly due to direct reduction by TrxR1 in the cellular context.

(2b) Selenium supplementation/depletion has no effect on TRFS-green signal generation in cells, even though this controls the amount of active site selenolthiol TrxR (new **Fig 5, new Fig S8**). Thiophosphate supplementation (new **Fig 5, new Fig S8**) which likewise prevents selenocysteine incorporation, is also ineffective at reducing TRFS-green signal. Unsurprisingly to us, SS50-PQ performs identically in both these respects i.e. is similarly nonresponsive to changes in the amount of functional cellular TrxR. We conclude again that cellular reduction of 1,2-dithiolane probes does not depend on correctly-expressed UC-containing TrxR. For comparison to how a TrxR-dependent probe should perform, in new **Fig 5** we now also reference results of our very recent work in selenium chemistry developing the redox probe candidate RX1, that data consistently indicates to be near-exclusively directly activated by TrxR1: e.g. the >90% signal suppression by TrxR1 knockout, the signal enhancement by selenium supplementation, and the signal reduction by thiophosphate treatment (further details in doi: 10.33774/chemrxiv-2021-52kwx).

(3) We also introduce new comparisons between several probes to show how assay interpretation requires combining independent experiments in order to advance; these settle any remaining questions about our statements in regard to previously published reports. For example: (3a) 1,2-dithiolane SS50-PQ is unaffected by cellular TrxR knockout (new **Fig 5**, new **Fig S8**): so it is not significantly cellularly activated by TrxR: yet its signal is mildly suppressed by cellular treatment with chalcophilic S_NAr -based electrophiles such as TRi-1 and TRi-3 (new **Fig 5**), and strongly suppressed by cellular treatment with the lipophilic Au (I)-based Lewis acid auranofin (AF), which has more than 20 attested targets (new **Fig S10-15**). The conclusion is that treating cells with these thiol/selenol-affine electrophiles and observing this suppression of cellular signal from a dithiolane probe, cannot be cited as a proof that that 1,2-dithiolane-based probe is a selective reporter of TrxR, since SS50-PQ provides a clear counterexample. We also must take into account e.g. Matile's studies, that showed that 1,2-dithiolanes substantially rely on *free* exofacial thiols for cellular uptake. This uptake is inhibited by general thiol-reactive species of all tested chemotypes (including even other disulfides) typically even suppressing uptake to only 10% of normal. This provides a coherent explanation that matches classical organic chemistry: these lipophilic electrophiles can suppress cellular 1,2-dithiolane signal generation by reacting with thiols on the cell surface, so substantially blocking the otherwise strain-promoted enhanced cellular uptake that dithiolanes can experience. (3b) Now we examine the electrophile treatment data for 1,2-dithiolane **TRFS-green**. This shows almost identical signal suppression as SS50-PQ (new **Fig 5**, new **Fig S11**). We have excluded that the electrophile assay tests *TrxR selectivity*: but, supported by literature, we would expect that these electrophiles should very similarly inhibit strain-promoted cellular uptake of the dithiolane TRFS-green as for the dithiolane SS50-PQ; and the observation that the level of inhibition is so similar between SS50-PQ and TRFS-green again suggests that their chemical behaviour (rooted in their 1,2-dithiolane) is the same, i.e. that the signal of TRFS-green is likewise being inhibited by electrophilic blocking of cellular thiols, not from the effects those electrophiles also have by partially reacting with cellular TrxR (which we stress, is only one of their many cellular targets). (3c) To counter-test these results, we now bring in the linear disulfide probe **SS00-PQ**. This probe should not benefit from strain-promoted thiol-mediated uptake enhancement (which can be suppressed by electrophiles); and we as well as others have shown that linear disulfide probes are not selective for any particular cellular reductant - while they are reducible by TrxR, they are also reducible by the vastly more concentrated GSH, Trx, Grx, etc. The electrophile assay results are perfectly coherent with this expectation (new **Fig 5**, new **Fig S12**): the electrophiles do not suppress signal from SS00-PQ, since the signal from this non-dithiolane probe is not limited by strain-promoted uptake.

This new intercomparison of results across multiple probes before reaching conclusions (new **Fig S13**) is, we feel, a strength of our paper and one which was not present in previously published reports; and given the unity of results between our releasing probe SS50-PQ and the alternative releasing design TRFS-green, we consider that this delivers solid evidence for the generality of the nonselectivity and effects that we report.

Secondly, we now supply cell-free experiments with the **non-releasing** 1,2-dithiolane-based probe Fast-TRFS (Fang et al. **2019 Nat. Commun.**, 10.1038/s41467-019-10807-8). We reached important conclusions only by our comparison to its novel linear disulfide analogue "Linear-TRFS" of which we also give the first report and characterisation (we include much of the following logic as expanded discussion in **Supplementary Note 3**). Fast-TRFS was intended to be fluorescence-quenched in its oxidised cyclic 1,2-dithiolane state, and was reported to be "a specific and superfast fluorogenic probe of mammalian thioredoxin reductase" that should act in cells by increasing in fluorescence upon "reduction [to Dithiol-TRFS]". We now show that there are at least three hitherto unreported confounding aspects to this Fast-TRFS probe system, and discuss how these combine with each other to give misleading results: and in doing so we support our thesis that 1,2-dithiolane is not a viable redox sensor for selective probes.

Firstly, as we discuss in relation to new **Fig 6**, in our hands, Fast-TRFS performs with poor reproducibility that matches entirely to Whitesides' characterisation of polymerisation-prone dithiolanes, and to the observations we report about polymerisation-prone SS50-PQ and its precursors.

Secondly, we show that a ring-opened species resulting from strain-promoted thiol attack on the Fast-TRFS 1,2-dithiolane [model compound Linear-TRFS] will be just as fluorescent, as the fully-reduced Dithiol-TRFS (e.g. on the dose-response panel in **Fig 6d**, at 9 mg/mL lecithin, ca. 36 fluorescence units). This means that any of several likely TrxR-independent ring-opening mechanisms will generate TrxR-independent fluorescence signal from Fast-TRFS (**Fig 6**, **Fig S16**). These mechanisms include but are not limited to: (a) strain-promoted oligomerisation, that should be particularly relevant if high local concentrations of Fast-TRFS are created, resulting in all ring-opened probe molecules becoming fluorescent (see below and shown in **Fig 6c**); (b) strain-promoted cellular uptake is via the known mechanism of exofacial thiol attack on the dithiolane and so will likewise activate fluorescence before it ever reaches the intracellular environment; (c) opening of the dithiolane by any cellular thiol.

Thirdly, we show that the fluorescence signal from Fast-TRFS and from its oligomerisation and its reduction products, also have strong environment dependence which creates additional problems. For example, oxidised Fast-TRFS fluorescence is instantly ca. 3-fold enhanced just by moving from aqueous to apolar environment (at the first minute of **Fig 6d**, the fluorescence intensity without vesicles is 0.99, but rises to ca. 3.4 as the lipid concentration increases - i.e. already 1/10 of the total potential fluorescence signal) - and this is before *non-reductive* strain-promoted ring-opening has significantly taken place (which it then does, reaching 50% of completion within the next 29 minutes). We also show that the environment-dependence of the Fast-TRFS system entirely rules the fluorescence behaviour of the model compound Linear-TRFS, which is up to 17-fold enhanced just by exposure to lipids (**Fig 6d**: from 2.0 to 35) but is *almost entirely unaffected by reduction* (irrespective of which environment the probe is in - see TCEP spike at the end of its incubation, **Fig S16**). Therefore, the faster that a Fast-TRFS probe of any redox status accumulates into (intracellular) lipid environments, the more quickly that the cellular fluorescence will rise.

The combination of these three effects provided us a unified model rationalising the published "superfast" fluorescence turn-on of Fast-TRFS, depending on the situation under testing.

We propose that totally TrxR-independent, but simply local-concentration-dependent ring-opening-oligomerisation can be activated by partitioning the 1,2-dithiolane Fast-TRFS from a relatively large volume of aqueous medium into a relatively smaller volume of apolar environment (ideally, of high surface area for rapidity). Then Fast-TRFS fluorescence would rapidly rise as increasing amounts of the more fluorescent oligomer would be formed. We predicted that this could allow even maximal theoretical fluorescence signal to be reached either with catalytic, or even zero, added reductants. We tested this experimentally with a series of vesicle assays (**Fig 6** and **Fig S16**), simply sonicating commercial soybean lecithin (Sigma, P5638, ca. 60% phospholipids and 35% oils; zero content of TrxR or NADPH) diluted to <1%wt in distilled water to form the lipid vesicles, and applying Fast-TRFS to various concentrations of these vesicles. Surpassing our expectations, in just one hour, the Fast-TRFS signal in 0.9%wt vesicle mixture without any reductants reached *full maximal signal plateau, corresponding to that seen with full TCEP reduction to Dithiol-TRFS*. This can only be understood as local-concentration-dependent oligomerisation to the poly-(disulfido-TRFS), that is just as fluorescent as Linear-TRFS or Dithiol-TRFS when they are compared in the vesicle system (**Fig S16**). This shows the confounding influence of redox-independent, strain-promoted processes on 1,2-dithiolane chemical behaviour: and illustrates how the environment-dependent readout of the Fast-TRFS compounds expands this complexity by generating a false positive signal.

Thirdly, as the aminonaphthylimide core of TRFS-green is also a classic environment-dependent fluorophore, and since its slow anilide elimination suggested that it could only perform fluorogenically in cells by harnessing environment-dependent effects that might well be independent of reduction (new discussions at **Fig S3**) we looked in the literature to see if environment-dependency might be shown as a similarly confounding issue elsewhere. We now added a discussion (**Supplementary Note 4**) to this effect. For

example, we note that recently published results from the Fang group (10.1021/acssensors.1c00049) show that their naphthilimide-based compound **S1**, which is similar to TRFS-green, experiences an instantaneous 45-fold enhancement of fluorescence intensity upon leaving all-aqueous environment and noncovalently associating to albumin in a cell-free experiment (their Figure 3; note too that their aminocoumarin compound **S3**, which is similar to Fast-TRFS, has a ca. 4-fold fluorescence enhancement in the same experiment, matching our vesicle results with Fast-TRFS).

Paralleling our experimental investigation of TrxR-independent signal generation in the Fast-TRFS system, this literature report of environment-dependent-signal highlights how interpreting fluorescence increases with the TRFS-green probe may similarly not be straightforward since (1) in any context, even just leaving the aqueous (extracellular) environment will trigger a fluorescence increase, that can be entirely independent of any reaction on its 1,2-dithiolane motif; (2) in the cellular context, all TrxR-independent strain-promoted thiol-mediated uptake at the cell surface will covalently associate the TRFS-green probe onto membrane proteins and into membranes, thereby giving cell-uptake-driven signal independent of molecular encounter of the TRFS-green probe with intracellular TrxR, let alone cyclisation-driven release of the cargo. It is therefore consistent with these hypotheses that the same factors as we advanced for the 1,2-dithiolane Fast-TRFS (concentration into membranes aided by membrane-thiol-based opening of the strained dithiolane, which gives an environment-dependent signal turn-on) will apply to TRFS-green, permitting it generate signal based on cellular exofacial thiol status, without even encountering TrxR. These suggestions should be considered in light of the demonstration (see above) that cellular signal from TRFS-green is manifestly independent of TrxR.

In conclusion: through the two new time-intensive biochemical experimental series with TRFS-green and Fast-TRFS, we consider that we have now gone far enough beyond our original scope of demonstrating the general liabilities of 1,2-dithiolane, fully responding to reviewer recommendations to illuminate how such entirely general chemical problems of 1,2-dithiolane can be amplified, according to chemical design, into strongly confounded readouts that may easily give misleading chemical interpretations. In doing so, we have tackled both cargo-releasing as well as non-cargo-releasing 1,2-dithiolane probe types. We feel that these results build in one paper, a larger and more convincing body of evidence than has hitherto been accessible even by combining multiple literature sources, and which matches both chemical logic as well as prior literature in a manner we feel is both convincing and coherent. There are still other literature-reported 1,2-dithiolane probes and even prodrugs, but our aim is to talk about strained disulfide 1,2-dithiolane probes in general, and not about specific probes or reports; and as we believe this now builds a very strong general case we consider this the right moment to stop.

2. The AF inhibition data is important. It is reasonable to assume that the probe interaction with AF would decrease its signal. But its not clear why the authors (apparently?) just hypothesized this rather than doing such a seemingly simple control experiment. Also, was AF inhibition the only control used in the prior work under challenge? Didnt they use any control disulfide-reacting species such as cysteine or glutathione as the current authors herein use? What were the prior results, how do they compare to the controls described herein?

Please note: "the authors (apparently?) just hypothesized this" refers to Fang et al., not to our paper.

Given the greater emphasis we introduced on Fast-TRFS and TRFS-green and on cellular studies, our treatment of catalytic, cell-and-enzyme-free, auranofin-mediated signal suppression by strain-promoted oligomerisation has had to take a smaller position in the revised manuscript. Nothing has been removed, and many months of experiments have been added (new **Fig S9-S15** and **Supplementary Note 2: Auranofin**): but we have consolidated all this data, discussion and references in the Supporting Information, with only a short summary and link to it in the Main Text.

We have tested the catalytic suppression of reducibility in several additional confirmatory experimental rounds, which supported our previous hypothesis that AF treatment can likely catalyse forming nonreducible, likely aggregated/precipitated oligo-SS50-PQ (new **Fig S9**). We appreciate the referee's

interest in AF and its role in previous studies; we also think this is an engaging feature that we can bring to the community's attention. However, we cannot go assay-by-assay-style through previous studies or speculate about the controls they chose. What we instead did is to run the cell-free AF assay on unstrained linear SS00-PQ and unstrained cyclic SS66C-PQ as comparison species, showing as expected that there was no inhibition of signal generation for either of these compounds, which supports the strain-promoted mechanism (new **Fig S9a**). With this added result, we feel that our expanded discussion, particularly in light of the extensive generalised investigations into the previously published 1,2-dithiolane probes detailed above, provides sufficient mechanistic insight to demonstrate a convincing case to the reader about the actual performance of our and previously published probes regardless of what controls were or were not used in other works; and to draw a line under it we refer the reader to the **further references related to AF-phosphine dissociation (Supplementary Note 2)**, which we hope addresses this point satisfactorily.

3. There are other published TrxR probes that have other seemingly nonspecific functional groups such as linear disulfides, including relatively simple commercial kits (Ellman's reagent?). Why would these other reagents not be commented upon by the authors herein?

We didn't comment, because our scope is showing the unsuitability of 1,2-dithiolane in cells. Therefore, our literature summary treatment (**Fig S2**) focuses on 1,2-dithiolane probes. We are also very interested developing our own novel, robust, cellularly selective probes, such as the RX1 probe for TrxR (10.33774/chemrxiv-2021-52kwx); but the proper place to deal with alternative TrxR probes is in that paper not here. Also: with a cap at 70 references, it would not be possible to tackle even a fraction of the different systems that have been claimed as relating to TrxR in this paper. And as most in the field are aware, many of those compounds like Ellman's reagent are clearly nonsense in the cellular context too - but it would not bring our probe research forward to discuss them here. Still, we appreciate the suggestion; we ask the referee's understanding that this paper is not the right place to lose focus on 1,2-dithiolane.

We thank Reviewer 1 for their close reading and carefully considered opinions on the paper, which challenged and encouraged us to dive a couple of levels deeper into 1,2-dithiolane biochemistry, and which we are sure now makes for a very much stronger paper.

Reviewer #2: The goal of this work was to verify whether the 1,2-dithiolane-based probes, claimed to be specific inhibitors of mammalian thioredoxin reductase, are indeed specific. To this end, the authors have designed a new probe based on previous work. In vitro assays clearly shows that monothiols such as cysteine (Cys), N-acetylcysteine (NAC), N,N-dimethyl-cysteamine (MEDA), and cysteamine (CA) and the reductive enzymatic systems – Trx/TrxR and Grx/GSH/GR are excellent reductant of the 1,2-dithiolane moiety and generate rapid fluorescence response. In other words, the probe is not specific.

We agree, but, particularly given the new results shown in **Fig 5-6**, we would rather rephrase it that the *1,2-dithiolane motif itself is not selective*, and that the probes we have tested this motif in, merely reveal this nonselectivity. We are more consistent about this now in the text. Additionally, we have now expanded our 1,2-dithiolane assessment with TRFS-green and Fast-TRFS, as well as key non-dithiolane comparison probes RX1, SS00-PQ, Linear-TRFS, and SS66C-PQ (see major discussions above). Through this combination of tests we are now sure that we demonstrate the general unsuitability of 1,2-dithiolane for "enzyme-specific" interpretation in the cellular context.

However, the fluorescence microscopy (Figure 4c), flow-cytometry (Figure 4d), and the fluorescence imaging of zebrafish (Figure 4e) don't add much to the story and may even be a distraction to some readers. Additionally, the animal study is complicated because of the expression pattern of TrxR1 at various life stages of the animal.

We feel that these experiments showing the quality of our PQ probe performance are a central part of our story (responding to **goal b**: suitability of general probe design). We have now rephrased passages to make this clearer. Our scope has been to show the non-selectivity of 1,2-dithiolanes (**goal a**), by using an unimpeachably robust and easily-interpreted environment-independent probe system that can be

extended to any arbitrary redox sensor trigger units, and can deliver valuable information by FACS, live animal imaging thanks to cellular retention, etc (**goal b**). We now clarified this in the text, also by citing our two more recent papers that rely on the cellular and *in vivo* performance that we have established and benchmarked in this work (10.1021/jacs.1c03234, 10.33774/chemrxiv-2021-52kwx). We did not intend the animal to complicate anything - we **now clarified that** since the 1,2-dithiolane probes are anyway not selective for any one reductant, the signal they provide cannot and absolutely should not be interpreted as being related to TrxR1 (which is expressed and active inside all somatic cells since it is needed for Trx maintenance and for DNA synthesis, etc). With these changes we feel that the images and proof of concept animal application clearly form a part of the coherent investigation.

Moreover, the determination of the mechanism by which AF inhibits the interaction between TrxR1 and the probe is critical and will add great value to this work. Whether or not AF's binding to membrane thiols actually affects the cellular entrance of the probe can be investigated by designing a similar probe in which the fluorophore is always turned on and remains attached to the 1,2-dithiolane moiety irrespective of its redox status. Also, the author should consider the possibility that when the probe enters the cell and gets reduced to the thiol form, it can react in the reduced form with AF, which, as the author mentioned, is a "potent thiol-reactive species". Expectedly, the progress of the reaction can be monitored using mass spectrometry or some other method.

We thank the reviewer for highlighting this. Indeed, we feel that bringing this incompatibility of AF with reducible strained disulfide probes to the attention of the redox community is an important feature arising from our work: even though as we wrote above, we have had to concentrate all the AF experiments into the Supporting Information to cope with the scope expansion in the Main Text.

We feel that now with the Supporting Information treatment of the AF experiments, expanded to stretch over **Fig S9-S15** and the expanded accompanying **Supporting Note 2**, this makes a self-complete demonstration to alert the community to this problem. We also now cite the relevant paper (10.1002/anie.201502358) at several relevant junctures, since it has employed exactly such a compound as the referee proposed (permanently-fluorescent xanthene attached to 1,2-dithiolanes: compounds **3** and **4** in that paper) for a similar purpose, and demonstrated similar dependency of net cellular fluorescence upon the redox status of exofacial thiols (which requires assuming a thiol-dependent uptake mechanism to be the ruling feature of cell entry for 1,2-dithiolanes, which in turn suggests just as we do, that 1,2-dithiolane cannot be interpreted as a TrxR-selective motif). We think in these ways that we have highlighted, and proposed rational explanations for, key problematic features that have never been raised before in this community. For example, the ligand exchange that permits the AF phosphine to initiate reduction of one dithiolane, which had been observed in coordination chemistry work, but had never been cited as an issue for redox biology (**Supporting Note 2**).

Like the reviewer we are interested to know what AF does. However, we do think that resolving the spectrum of things AF *actually* does, and under what circumstances, is a daunting topic even split over many research groups - it is no accident that after thousands of papers using it, a typical summary of AF's effects remains "we still do not understand the scope and consequences of AF's cellular targets". AF is known to be a dirty drug, and the sooner our group stops investigating its drawbacks to focus exclusively on reducible probes that are used in clean assays and with clean inhibition strategies, the better. Deceptively simple though such an experiment might appear, the complexity of the cellular membrane composition, lack of good species-resolved analysis techniques, and especially the lack of a good simplified *in vitro* model make this a tricky challenge. Furthermore, we only treat auranofin as an example of one way in which lack of cross-disciplinary communication and appropriate controls, has probably misled previous investigators - and we have to orient our investigations around 1,2-dithiolane in this paper.

Regarding probe-AF interaction inside cells, we were pleased to get this remark. We now added to our Supporting Information discussion (end of **Supporting Note 2**) that we think that the intracellular background thiol concentration (5mM GSH, up to 50 mM protein thiols) is so much higher than the instantaneous reduced intracellular probe reduction dithiol intermediate is ever likely to be (particularly for the rapidly-cyclising SS50-PQ), that since we see no basis for preferential AF reaction of the reduced dithiol

intermediate compared to any other monothiol, we do not imagine that intracellular reaction can be significant to suppression of signal generation.

In conclusion, by bringing this, as well as the polypharmacological targets of AF to light in this community, we feel that we have delivered substantial advances that will suitably caution or inspire redox biologists in future studies. As we feel that our several figures of AF results are already proportionally of greater weight than they should be given that our paper focuses on 1,2-dithiolane chemistry and on probe performance criteria, we fear that diving more into the polypharmacological AF's effects would explode the scope of this paper and distract from the chemical achievements that we have anyway shown: nonselectivity of reductants, and non-reduction-based signal generation, in cellular and in cell-free tests. Therefore, we trust that the revised manuscript now addresses these issues at a suitable level for publication.

Other Major Changes:

- Synthesis, characterisation, cell-free and cellular experiments with literature known compound TRFS-green (Fang et al. JACS 2014), shown in Figure 3, Figure 4 and Figure 5, extended in Supporting Information Fig S3 and Fig S5, as well as Fig S7, Fig S8 and summarized in Fig S1.

- Extended evaluation of TRi and AF inhibition of cellular processing of SS50-PQ and TRFS-green and intercomparison to a negative (SS00-PQ) and a positive (RX1) compound, shown in Figure 5, extended in Supporting Information Fig S11, Fig 12 and Fig S13, summarized in Fig S1.

- Synthesis, characterisation, and cell-free vesicle experiments with literature known compound Fast-TRFS (Fang et al. Nat Commun 2019) and with our novel non-strained linear disulfide model compound Linear-TRFS: vesicle data Fig 6 & Fig S16, short discussion in main text & expanded discussion in Supporting Information related to Fig S16.

- New Figure S1 has been provided (summary overview), to assist the reader in comparing the results in this work which pertain to TrxR selectivity or lack thereof.

=====

REVIEWER COMMENTS

Reviewer #1 (Remarks to the Author):

This revised manuscript includes additional experiments to augment the authors' claims that certain cyclic 1,2-dithiolanes in general, and the known dithiolane TRFS probes specifically, are not selective for TrxR and should not be used as probes for TrxR. The paper is focused on proving significant interpretations of prior, peer-reviewed work are wrong.

To address the prior review comments, the authors synthesized and studied TRFS probes that they are criticizing. This is commendable. However, the newly reported findings are still too speculative for consideration for publication in a top-tier journal.

For example, one proposal is that the cyclic dithiolanes polymerize and that this causes non-specific fluorescence that interferes with TrxR signaling. However, the experiments were not performed in cells containing TrxR, and, though a signal was generated in vesicles, there is no direct proof of any polymerization event given, despite the experiment in (more tractable) artificial media. The knockout experiments show that a potentially troubling non-specific signal is formed by the TRFS probe in cell media. However, without the presence of TrxR, this does not take into account, or directly demonstrate, that the cellular reaction with TrxR may have superior kinetics compared to whatever mechanism is causing the signal under knockout conditions. While the authors' suppositions may have validity, that reactivity with other biomolecules apart from TrxR does not allow the TRFS and related probes to function properly, they have not provided strong enough evidence to make such claims.

In the prior review, the suggestion was made to consider an analogy with covalent drugs. I will be more specific here. Fosfomycin, for example, contains an electrophilic epoxide. Worse, Zanubrutinib contains an acrylate moiety!

Using the authors' rationale, none of these (or the numerous other covalent inhibitors that are now major prescribed pharmaceuticals), should have ever been marketed, as it is so easy to show that their functionality exhibits well-known covalent reaction promiscuity, that these authors would argue ensures that none should never exist long enough to reach the desired protein target.

One could argue, as the researchers here have done, for example, via extensive chemical literature citations about basic chemistry of such functionality, and one could readily design several experiments in non-natural model conditions, to "prove" that such drugs do not perform as claimed, because they react with other biomolecules besides their targets.

Finally, the paper and the responses are relatively dense. This is not meant as a criticism of the paper or the writing style. The broader issue is that a non-specialist reader of a multidisciplinary journal can likely be overwhelmed by the extensive arguments and citations that can obscure the lack of substantive experimental proof and the lack of a balanced assessment of the prior probe modalities criticized herein.

In summary, the authors afford ample proof that the compounds and functional groups they are questioning are indeed reactive with species besides TrxR. However, this is already a well-known fact, that these functional groups are highly reactive. Moreover, they have not shown compelling, direct substantive evidence to overturn peer-reviewed results showing the efficacy of the TRFS and related TrxR probes.

Reviewer #2 (Remarks to the Author):

All reviewer comments have been satisfactorily addressed

=====

Reviewer 1

This revised manuscript includes additional experiments to augment the authors' claims that certain cyclic 1,2-dithiolanes in general, and the known dithiolane TRFS probes specifically, are not selective for TrxR and should not be used as probes for TrxR. The paper is focused on proving significant interpretations of prior, peer-reviewed work are wrong.

Yes, our manuscript contains many conclusive additional experiments showing our message that 1,2-dithiolane probes should not be used as probes for cellular TrxR activity (particularly Fig 5).

We disagree that the paper is "focused on proving significant interpretations of prior, peer-reviewed work are wrong" - our paper really focuses on the 1,2-dithiolane system in general, showing its nonspecific lability to ring-opening by monothiols or by partitioning-based bioconcentration; *and* we also show that these features *do* prevent it from delivering information that can be interpreted as quantifying TrxR activity in the cellular context when the dithiolane operates a fluorogenic probe; but the focus is really on the motif (not just on specific incarnations of it in prior art probes), and how it actually behaves (not just what it cannot do).

The additional experiments and clarifications we have provided, show five orthogonal cellular proofs of this nonspecific reactivity: not only for the general dithiolane probe SS50PQ but now additionally for the debated dithiolane TRFS probes, in major part because Reviewer 1 requested that we add such experiments (Fig 5); and they also showcase interesting reduction-independent behaviour of Fast-TRFS which provides a seamless link to the extensive materials science literature on non-reductive templated 1,2-dithiolane ring-opening polymerisation (Fig 6) which we cite purposefully in the text.

We believe these results have the breadth and rigour needed to make a valuable step in the chemical biology of redox probes, by orienting future researchers towards valuable avenues for 1,2-dithiolanes to contribute meaningfully to scientific progress; these avenues do not include use as TrxR probes.

To address the prior review comments, the authors synthesized and studied TRFS probes that they are criticizing. This is commendable. However, the newly reported findings are still too speculative for consideration for publication in a top-tier journal. For example, one proposal is that the cyclic dithiolanes polymerize and that this causes non-specific fluorescence that interferes with TrxR signaling. However, the experiments were not performed in cells containing TrxR, and, though a signal was generated in vesicles, there is no direct proof of any polymerization event given, despite the experiment in (more tractable) artificial media. The knockout experiments show that a potentially troubling non-specific signal is formed by the TRFS probe in cell media. However, without the presence of TrxR, this does not take into account, or directly demonstrate, that the cellular reaction with TrxR may have superior kinetics compared to whatever mechanism is causing the signal under knockout conditions. While the authors' suppositions may have validity, that reactivity with other biomolecules apart from TrxR does not allow the TRFS and related probes to function properly, they have not provided strong enough evidence to make such claims.

The most important point that the reviewer has objected to here and in the previous round, is whether sufficient evidence exists *despite* our paper to conclude that 1,2-dithiolane probes are selective cellular TrxR reporters, or whether in light of our paper sufficient evidence exists to show that they are *not*.

We provide primary experiments in cells and cell-free and using several probes, as well as literature rationale and references, that constitute multiple evidence types showing that they are *not*. We argue that these data in themselves are sufficient to state that the null hypothesis (dithiolane probes are not selective) is strongly supported. The reviewer may see the burden of proof differently; but choosing to believe certain prior claims about TrxR-selective reporting of dithiolanes *is also a choice* to disbelieve not only our primary experiments that shows they are not selective (see points 1-2 below), but additionally to disbelieve all the substantial literature that we for the first time collect together, showing that 1,2-dithiolanes are in no way TrxR-selective (see points 3 and 5). We believe either of those sets of data and experiments (ours or previous literature) should be sufficient to show nonselectivity beyond any reasonable doubt. And we regret that the reviewer would, for example, consider the concordant findings of five orthogonal experiments on exactly the TRFS probes that the reviewer requested (Figure 5) to be only "*speculative*" rather than meaningful. To clarify several major points, we highlight:

(1) We show that cellular knockout of TrxR does not abrogate even half the dithiolane/TRFS probe signal (Fig 5c). This is the first proof experiment that the TRFS probe simply cannot be quantified and interpreted as a cellular TrxR reporter - *it lights up in cells, but it does so when TrxR is not even there*.

We do not see how this can be ignored or deemed "*lack of substantive experimental proof ...or compelling, direct substantive evidence*".

We would also like to clarify: "*The knockout experiments show that a potentially troubling non-specific signal is formed by the TRFS probe in cell media*" - the knockout experiments actually show, that between two thirds up to the entire signal of fluorogenic dithiolane probes in cellular assays, is *conclusively not related* to TrxR (Fig 5c). We cannot imagine what stronger proof that 1,2-dithiolane probes are *not* cellular TrxR quantifiers could be shown. We would like to clarify another misperception: the signal in the knockout experiments is not formed in the *media*, it is produced by the *cells* (compare no-cell control assays throughout the text). This means conclusively that cells themselves turn on from 60-100% of baseline dithiolane probe signal unrelated to whether they have TrxR. We consider that this is easily strong enough evidence to show that TRFS and related probes are simply not functioning as claimed!

(2) There are still four further experiment types shown in Figure 5. Compare the two central columns (dithiolane probe results) of Fig 5a, Fig 5b, Fig 5d, and Fig 5e - in cells, these probes are manifestly not reporting on TrxR since they do not alter behaviour as a TrxR reporter should. They do not either gain or lose signal upon modulation of the levels of functional TrxR (Fig 5a-b); they did not respond to knockout (see point 1 above); and they are barely affected by potent TrxR inhibitors (Fig 5d-e). Now compare them to linear disulfide in column 1. As reviewer 1 correctly pointed out in review round 1, linear disulfides are not selective for TrxR; however its results are substantially identical to the dithiolane probes. Conclusion is: the dithiolane probes are exactly as non-TrxR-selective-reporters as the linear probe to which they perform identically across *five orthogonal cellular assays*. (For further comparison, see the final column of Fig 5,

which shows how a TrxR-selective reporter should (and does) behave - strong signal suppression by TrxR inhibition or knockout or suppression - very different from TRFS of SS50PQ dithiolanes).

We do not see how this can be ignored or deemed "*lack of substantive experimental proof ...or compelling, direct substantive evidence*".

(3) Still more "*substantive evidence to overturn peer-reviewed results [about TRFS being supposedly cellularly specific]*" is shown across all the papers we cite: from Whitesides in the 1980s through to Matile in the 2020s - please consult our references 20-43 as well as the Matile references cited above. What is new is that we collect this evidence in one relevant place and summarise it for the redox biology community for the first time. Perhaps this compelling evidence has caused a shock? If so, it highlights the need for our paper for this community; if not, then the reviewer should not disagree with our conclusions.

(4) We provide direct evidence about some cellularly relevant factors, whose significance and truth have unfortunately been resisted by certain parts of the community until now. One example that is crucial is that we show that structure-activity relationships *do* matter for the biological interactions of reducible motifs, just as much as they do for any other compounds: our evaluations across the different probes SS50PQ and TRFSgreen, sharing the same reducible motif, shows that the *same results are obtained* for them (which had been doubted by the reviewer in the first round of reviews). For example, Fig 3c-d shows the same off-target scope for both probes, just with different kinetics reflecting their leaving-group character.

Especially in light of the 23 studies we cite and the references within, we do not see how this can be ignored or deemed to be "*lack of substantive experimental proof ...or compelling, direct substantive evidence*" that dithiolanes systematically have the behaviour we describe.

(5) Finally, we would like to refer the reviewer to the body of literature also bringing dithiolane nonspecificity to light, which has exploded in the 13 months since our paper entered review, and which can help inform a balanced view of strained disulfides. For example, the recent comprehensive Matile review "Thiol-Mediated Uptake" (JACSAu 2021, copy attached, now cited in our manuscript) highlights this exhaustively: (i) 1,2-dithiolane **5** in that work is a "milestone thiol-mediated uptake" motif (Fig 2), whose nonspecific monothiol polymerisability (**8**, in Fig 13), reflects its "speed... of dynamic covalent exchange chemistry *on the way into cells*" (section 3.2 and Fig 9a, Fig 14, and **C** in Fig 15) - i.e. *prior to any possible exposure to the cytosolic enzyme TrxR*. (ii) That same review lists multiple extracellular protein targets of dithiolanes (section 5.7), from TFRC to EGFR to PDIs - again, all of which reflect that dithiolanes are not simply targeting and reporting on cellular TrxR, but are already opened by exofacial thiols that they encounter before even getting into the cell. (iii) for reviewer interest, we note that Matile's review consistently highlight's dithiolane's behaviour as a *dynamic covalent* motif (see Figures cited above); this may clarify why the reviewer's questions about irreversible electrophilic epoxides and other irreversible covalent drugs are additionally not relevant to this paper. We are sure that the reviewer will find many other such experimental references in those sections of the review and in the accompanying references, which support our interpretation that dithiolanes are not cellularly selective TrxR inhibitors, but are nonspecifically and even nonenzymatically opened to form polymers (Fig 14), even templated in vesicles (Fig 15) - the aspects that our experiments and text have likewise maintained but which the reviewer may desire additional outside confirmation about.

Taking these five aspects together, we do not see how they build a position that can be ignored or can be deemed a "*lack of substantive experimental proof ...or compelling, direct substantive evidence*".

In the prior review, the suggestion was made to consider an analogy with covalent drugs. I will be more specific here. Fosfomycin, for example, contains an electrophilic epoxide. Worse, Zanubrutinib contains an acrylate moiety!

Using the authors' rationale, none of these (or the numerous other covalent inhibitors that are now major prescribed pharmaceuticals), should have ever been marketed, as it is so easy to show that their functionality exhibits well-known covalent reaction promiscuity, that these

authors would argue ensures that none should never exist long enough to reach the desired protein target.

One could argue, as the researchers here have done, for example, via extensive chemical literature citations about basic chemistry of such functionality, and one could readily design several experiments in non-natural model conditions, to "prove" that such drugs do not perform as claimed, because they react with other biomolecules besides their targets.

These paragraphs are not relevant to evaluating the science we present (dithiolanes, not irreversible-covalent drugs). We prefer to focus on relevant facts and refer the reader back to points 1-5 which deal with relevant facts. (Although, in passing, we are sure that if Fosfomycin and Zanubrutinib were shown to be activated at 60-100% of normal rates when Protein A is deleted from cells, the reviewer would agree that Fosfomycin and Zanubrutinib were thus shown not to be selective reporters of Protein A...?)

Finally, the paper and the responses are relatively dense. This is not meant as a criticism of the paper or the writing style. The broader issue is that a non-specialist reader of a multidisciplinary journal can likely be overwhelmed by the extensive arguments and citations that can obscure the lack of substantive experimental proof and the lack of a balanced assessment of the prior probe modalities criticized herein.

In summary, the authors afford ample proof that the compounds and functional groups they are questioning are indeed reactive with species besides TrxR. However, this is already a well-known fact, that these functional groups are highly reactive. Moreover, they have not shown compelling, direct substantive evidence to overturn peer-reviewed results showing the efficacy of the TRFS and related TrxR probes.

We do not understand how on the one hand, we can be said to provide "ample proof", how we "synthesized and studied TRFS probes that [we] are criticizing [which] is commendable", and yet elsewhere the reviewer states there is a "*lack of substantive experimental proof and the lack of a balanced assessment of the prior probe modalities criticized herein*". We also do not find it consistent that the reviewer says in one place "this is already a well-known fact, that these functional groups are highly reactive" and yet doubts whether dithiolanes can, on the basis of our or any other data, even be *expected* to have off-target reactivity:

We feel that our five-point summary (1-5) outlined above shows compelling, direct substantive evidence from us as well as from a substantial body of literature, which should easily overturn any convictions that TRFS and related TrxR probes *could* be selective cellular reporters of TrxR, no matter what prior convictions or exposure to the field a researcher may have; and we request that the reviewer reconsider their opinions of our work and our cited literature accordingly.

As to the density of the manuscript, much of the data and references are required in order to keep up with reviewer's requests or to provide solid information: there is not much to be done there.

In fact, many literature precedents have appeared since our submission also supporting that 1,2-dithiolanes are not TrxR-selective reporters. We still feel that our paper is still a unique milestone towards correcting and reconciling interpretations of dithiolane cellular activity. Just as Reviewer 1 has maintained doubts in the face of the evidence that our paper brings against previously published *interpretations* of experimental results (note: we have never disagreed with the experimental results themselves), so doubtlessly have others: so we feel our paper is a very timely one.

Ours is the first paper to openly tackle these chemical biology probe reports that use dithiolane as if it were a cellularly selective TrxR reporter. We take the time and effort to experimentally revisit the previously published probes and investigations, and to expand on them by performing necessary controls that allow us to elucidate how they may have fallen into mistaken interpretation (or which controls have been missing). Therefore we believe that finally bringing this paper to the community is of very urgent value to the field. We feel more than ever that, the reviewer's initial doubts notwithstanding, the increasingly accruing evidence

also from other groups' research that our experiments have been robust and our interpretations have been correct, render our paper a timely and valuable addition to the literature.

Reviewer #2

All reviewer comments have been satisfactorily addressed

We thank the reviewer for a frank and clear assessment of the paper and our replies to reviews.

=====

In conclusion, our work has examined both novel and literature-known dithiolane probes, to provide multifaceted experimental evidence for a coherent model of their cell-free and cellular chemistry that both corrects the current state of the literature, and orients future redox biochemistry research in new directions. We believe that this in-depth study brings a range of unique and timely contributions to the field, and will prove a valuable stepping stone in the chemistry and biochemistry of cellular oxidoreductases.

We therefore consider that this revised and improved paper will achieve high impact with a broad audience of researchers from chemical biology, to redox biology, biochemistry, and chemical probes. We particularly expect that our interdisciplinary article will reach out to a broad Nature Communications readership, and we look forward to your feedback and comments.

The manuscript is not under consideration for publication and has not been published elsewhere. We filed a preprint on ChemRxiv late 2020 and updated it to reflect the version resubmitted with this letter.

REVIEWER COMMENTS

Reviewer #1 (Remarks to the Author):

These researchers are to be commended for the effort they have put into this detailed study. Further, they have been proven correct (by the authors of the original art) in their assessment of the TRFS probes, the probes they had criticized (expressed concern over) since the first iteration of these submissions. I also agree that the additional experiments they had added upon my prior request(s) aided the case for the manuscript.

I still have some concerns, however, about the sometimes overly-speculative nature (relevance) of some, not all, of their conclusions, as stated in my prior review(s). Unfortunately, the authors have misrepresented some points made previously:

It is not a choice of mine to believe certain facts and not others, (see "rebuttal" file, "null hypothesis" comment). The dithiolane probes are indeed generally chemically non-selective. Anyone who does not see that is scientifically blind. I never doubted that. I never claimed that in the prior review cycles.

The point in my critique(s) was whether they are non-selective enough under the exact same conditions (to be fair to the prior research groups), used and reported by the prior researchers whose work was being questioned.

Too many papers challenge prior art but only after working in a different experimental context than was initially intended or used by the original researchers, and inappropriate conclusions are made.

Moreover, no probes or drugs have perfect selectivity. Selectivity is condition and matrix dependent, among many other factors, especially in biological milieu, of course. It is with these issues in mind that I continue to address/have to clarify the points made previously.

To challenge (especially prior peer-reviewed) work without repeating the prior work, and in a manner that is very clearly performed and presented as a very rigorous replicate to the reader is not preferable.

For example, I was very happy to see the authors, after the first review round, decide to actually synthesize and use the same probe as was reported previously by those they are questioning. This went a long way in validating comparisons. I had suggested this in my initial review.

However, instead, the rebuttal claims "SS50PQ and TRFSgreen share the same reductable motif.....shows same results.....had been doubted by the reviewer."

To use just two probes to conclude that only the dithiolane group, and not the protein etc binding of the rest of the probes, is significant for selectivity, was a reach.

In addition, there was only doubt because the authors had tried to discount the prior work via initially making and using only a different probe than the earlier researchers had used. This was a suggestion of mine to only help strengthen the paper (i.e., to use an actual TRFS probe as a control), which it indeed did, and not due to "doubt" as has been attributed here.

I did not discount the fact that the probe lit up in the knockout case. I did not misinterpret this as stated in the "rebuttal". What I stated as a caveat, to add rigor, is that one must consider that the potentially interfering optical turn-on effects might not be kinetically competitive with the TrxR-derived signal. It is not due to some misunderstanding on my part. If that point is already in the paper or the rebuttal, it is not clear to me. Bottom line: one can't claim precisely a "60-100 % non-selectivity" without such a caveat.

"Perhaps this compelling evidence has caused a shock" is neither a helpful nor a professional comment. In fact, I noted in my critique that very many issues about selectivity of dithiolanes were already known (and published). That this fact was needed for background, but not new at all new and not surprising (please see my actual written critique to accurately represent and respond to my suggestions/concerns, thank you).

Again, the fact that the non-selectivity of this functional group is already well-known under other conditions in the prior art, was never the main issue for me.

The problem is that the exact experimental context of the TRFS etc dithiolane probe papers being criticized were not accounted for in a clear enough manner in the submission. Instead, carefully obtained, but general evidence for non-selectivity has been obtained and included.

Despite the current claims that the paper is not a critique of the prior work of others, that's exactly what it is (and was previously), but its at least good that the authors now at the very least try to claim that its only about the dithiolane.

Again, to be really clear a main point is that the authors need to show that, under the exact conditions of the prior art probe studies, very clear proof that they could not repeat the prior results and come up with the same conclusions.

This was only partially demonstrated, and/or not very clearly described in the manuscript. Now, however, the authors of the original prior art have shown that their prior dithiolane probe is indeed non-selective, and that their conclusions were wrong, under their conditions.

So it turns out that the authors of this manuscript were correct, though the way they "reproduced" (or didnt reproduce) the prior art using dithiolane probes, (much beyond making the same probe, but only did so upon the first revision), to make their point was, from the beginning, in question.

Apologies for the repetitive nature of aspects of this review, but I am doing my very best to be clear and understood here, since ts clear to me after reading the rebuttal that several issues raised previously were neither understood nor even carefully read. And I am still not certain of this, especially upon seeing the tone of some of the rebuttal comments.

However, I sincerely hope the comments (which are largely clarifications of my prior critiques) are helpful to the authors.

Taking all of the aforementioned into account as a whole, I overall support publication of this paper with minor concerns.

Reviewer #3 (Remarks to the Author):

I congratulate the authors for an excellent and courageous paper.

Reviewer #3 was asked to act as an arbitrator and see if reviewer #1's concerns have been addressed by the authors. They made comments to the editor and support publication.

After three rounds of detailed, constructive and collegial reviews/rebuttal we have addressed all reviewer concerns and we are confident that our manuscript is ready for acceptance and publication.

=====

Reviewer 1

These researchers are to be commended for the effort they have put into this detailed study. Further, they have been proven correct (by the authors of the original art) in their assessment of the TRFS probes, the probes they had criticized (expressed concern over) since the first iteration of these submissions. I also agree that the additional experiments they had added upon my prior request(s) aided the case for the manuscript.

We thank Reviewer #1 for the assessment. We furthermore thank them for their willingness to engage with our paper again and for clarification of their issues of concern. Our initial design had been to *not* resynthesise or directly critique previously published probes based on 1,2-dithiolanes but to show general dithiolane liabilities with an (in our opinion more easily quantified) probe SS50PQ. Their push to benchmark our results against actual prior-art probes has provided still more general proof, that chemotype-driven redox reactivity is indeed responsible for the liability in of dithiolane trigger-cargo-constructs regardless of appended chemical cargos.

I still have some concerns, however, about the sometimes overly-speculative nature (relevance) of some, not all, of their conclusions, as stated in my prior review(s). Unfortunately, the authors have misrepresented some points made previously: It is not a choice of mine to believe certain facts and not others, (see "rebuttal" file, "null hypothesis" comment). The dithiolane probes are indeed generally chemically non-selective. Anyone who does not see that is scientifically blind. I never doubted that. I never claimed that in the prior review cycles. The point in my critique(s) was whether they are non-selective enough under the exact same conditions (to be fair to the prior research groups), used and reported by the prior researchers whose work was being questioned. Too many papers challenge prior art but only after working in a different experimental context than was initially intended or used by the original researchers, and inappropriate conclusions are made. Moreover, no probes or drugs have perfect selectivity. Selectivity is condition and matrix dependent, among many other factors, especially in biological milieu, of course.

Yes, we can believe that papers can make challenges unfairly, and we share the reviewer's concerns to establish a just and accurate literature. We are convinced that our paper is not drawing inappropriate conclusions. We have seen the central issue as we present it (**highlighted in the introduction**): "*Which is the real situation? 1,2-dithiolane cannot be both highly and nonspecifically reactive, yet also TrxR-selective*

in the cellular setting," and we do think that there is a clear choice to be made between the "nonspecifically cellularly reactive" viewpoint, and the "cellularly TrxR-selective" viewpoint, with no real middle ground. We apologise for misunderstanding the reviewer's opinion on this matter and are glad that we share a scientific understanding. We really do not believe that we have unfairly attacked these probes (e.g. that they could have originally been used in ways that *were* reporting cleanly on cellular TrxR, but now, that we and their original authors have managed to find conditions where this is *not* true) - instead, we performed a great number of standard, logical, orthogonal assays, over a variety of conditions, and while we have never debated the accuracy of the previously performed *experiments* we do find that only the "nonspecifically reactive" viewpoint gives a coherent *interpretation* of the data available combining all sources (us+lit).

As to selectivity, of course, you are right. We don't even think there is an accepted numerical threshold separating "selectivity" from "non-selectivity" in the biological context (which makes it a slippery concept to refute). However, we feel that our results and those recently published by Fang (**new reference 53, discussed now in our paper at the highlighted Introduction / Conclusions parts**) establish that such dithiolanes really cannot be usefully interpreted as TrxR activity probes in cells; and given that these probes are the most widely-available ones being recommended, published, sold, and used for cellular TrxR, we think it is urgent to make the data underlying this interpretation of nonselectivity available to the community.

It is with these issues in mind that I continue to address/have to clarify the points made previously. To challenge (especially prior peer-reviewed) work without repeating the prior work, and in a manner that is very clearly performed and presented as a very rigorous replicate to the reader is not preferable. For example, I was very happy to see the authors, after the first review round, decide to actually synthesize and use the same probe as was reported previously by those they are questioning. This went a long way in validating comparisons. I had suggested this in my initial review. However, instead, the rebuttal claims "SSS0PQ and TRFSgreen share the same reductable motif.....shows same results.....had been doubted by the reviewer." To use just two probes to conclude that only the dithiolane group, and not the protein etc binding of the rest of the probes, is significant for selectivity, was a reach. In addition, there was only doubt because the authors had tried to discount the prior work via initially making and using only a different probe than the earlier researchers had used. This was a suggestion of mine to only help strengthen the paper (i.e., to use an actual TRFS probe as a control), which it indeed did, and not due to "doubt" as has been attributed here.

We thank the reviewer for their clarification and apologise for misunderstanding their review remarks. It was indeed a valuable suggestion to resynthesize and expand the datasets available around the originally-reported probes and this led directly to some interesting findings including those in **Fig 6**, thank you for the impulse to do so. Furthermore, we have been glad to add our data to the idea that redox structure-activity relationships of reducible triggers are determining features of the performance of trigger-cargo constructs within this space: which, we note, matches the logic of all published papers of TRFS dithiolanes that we cite (cationic and neutral, aromatic or nucleoside cargos: but same trigger: and thus same selectivity as each other); as well as all published papers in the thiol-mediated uptake field that **we now cite (refs 27-37)**. This large body of results around 1,2-dithiolanes that we now cite in one place for the first time, gives much confidence that the data we observe, by matching literature observations, are reliable general indicators.

I did not discount the fact that the probe lit up in the knockout case. I did not misinterpret this as stated in the "rebuttal". What I stated as a caveat, to add rigor, is that one must consider that the potentially interfering optical turn-on effects might not be kinetically competitive with the TrxR-derived signal. It is not due to some misunderstanding on my part. If that point is already in the paper or the rebuttal, it is not clear to me. Bottom line: one can't claim precisely a "60-100 % non-selectivity" without such a caveat.

This has been easy to clarify in our present, revised manuscript (p6-7 highlights). Yes, they are kinetically competitive. This is why we are confident in our interpretation. In our reply to 2nd round reviews where we specified, "We show that cellular knockout of TrxR does not abrogate even half the dithiolane/TRFS probe signal (Fig 5c)," we wished to show that these signals are acquired under as identical experimental conditions as can be achieved: same sampling times, same three probe concentrations

(25/50/100 μM), same cell seeding densities, etc - see also **Fig S8c** referenced from there. Since these signals are acquired under the same conditions, we plotted them on the same graphs (**Fig 5c**, columns 2 and 3), intending that the reader compare them quantitatively. To respond to this point we **now specified in the figure legend that any lines plotted on the same graph are intrinsically comparable (see legend to Fig 5)**. By comparing them, and finding no "black and white" difference, we concluded that the non-TrxR activation is indeed kinetically competitive, hence our conclusions. Thanks for the opportunity to clarify.

"Perhaps this compelling evidence has caused a shock" is neither a helpful nor a professional comment. In fact, I noted in my critique that very many issues about selectivity of dithiolanes were already known (and published). That this fact was needed for background, but not new at all new and not surprising (please see my actual written critique to accurately represent and respond to my suggestions/concerns, thank you). Again, the fact that the non-selectivity of this functional group is already well-known under other conditions in the prior art, was never the main issue for me. The problem is that the exact experimental context of the TRFS etc dithiolane probe papers being criticized were not accounted for in a clear enough manner in the submission. Instead, carefully obtained, but general evidence for non-selectivity has been obtained and included. Despite the current claims that the paper is not a critique of the prior work of others, that's exactly what it is (and was previously), but its at least good that the authors now at the very least try to claim that its only about the dithiolane. Again, **to be really clear a main point is that the authors need to show that, under the exact conditions of the prior art probe studies, very clear proof that they could not repeat the prior results and come up with the same conclusions**. This was only partially demonstrated, and/or not very clearly described in the manuscript. Now, however, the authors of the original prior art have shown that their prior dithiolane probe is indeed non-selective, and that their conclusions were wrong, under their conditions. So it turns out that the authors of this manuscript were correct, though the way they "reproduced" (or didnt reproduce) the prior art using dithiolane probes, (much beyond making the same probe, but only did so upon the first revision), to make their point was, from the beginning, in question. Apologies for the repetitive nature of aspects of this review, but I am doing my very best to be clear and understood here, since its clear to me after reading the rebuttal that several issues raised previously were neither understood nor even carefully read. And I am still not certain of this, especially upon seeing the tone of some of the rebuttal comments. However, I sincerely hope the comments (which are largely clarifications of my prior critiques) are helpful to the authors. Taking all of the aforementioned into account as a whole, I overall support publication of this paper with minor concerns.

We see that the reviewer writes that we would *"need to show that, under the exact conditions of the prior art probe studies, very clear proof that they could not repeat the prior results and come up with the same conclusions"*. But we just do not agree that this suggestion is helpful here; please let us explain why.

We did not and do not dispute many of the **experimental results** that are to be found in those prior papers - some of which we reproduced, others of which we don't believe we need to retest. But as the prior art probably reports 100% factually correct experimental results, re-testing them is not going to bring anything new, that's why we won't do more of it - it is not addressing the key problem, which is this:

What we did dispute was whether the **interpretation** "TrxR-selective" could be consistent with all data, considering thiol-mediated uptake and templated polymerisation. What those prior papers are missing are tests that could have probed selectivity: and therefore missed opportunities to test their **interpretations**. For example, in the Fang JACS 2014 paper reporting TRFS-green, there are no experiments testing TrxR knockout / knockin / selenium supplementation / selenium starvation in cells, and no screen of other thiol proteins such as Grx / Trx; the only evidence offered in support of TrxR-selectivity in cells was simply that 40 μM DNCB treatment (a broadly reactive chalcophilic electrophile) reduces cellular fluorescence signal.

Our contribution was that we greatly expanded the available set of results by designing new experiments to test whether cellular TrxR selectivity is likely: the knockout / knockin / auranofin / TRi / selenium supplementation / selenium starvation / Grx screen / Trx screen / new, more reliably quantified, environment-independent SS50PQ probe / new LinearTRFS control... This expanded set of results enables us to draw better-supported conclusions (**highlighted p12**). To continue the example, through these new experiments we have shown since the 1st round of revisions that TRFS-green is broadly sensitive to Trxs and Grxs and barely responds to TrxR modulations. And in this case, yes - the original authors have recently

revisited TRFS-green and now also state that it is a reporter on Trxs and Grxs: a conclusion that could neither be accepted nor denied just on the basis of repeating the originally reported assays.

More valuably, it is through such new experiments and methodology that we have been able to show, that a coherent understanding of dithiolane behaviour (nonspecific reactivity, manifested in different ways according to experimental circumstances) is also able to explain prior observations which had been (mis)interpreted as indicating TrxR-selectivity; and we believe our observations and also interpretations will be shared more widely by the community as our methodology becomes more widespread.

But to be clear: **supporting our interpretation does not require us to show** that if we repeat assays done previously we would come up with different results on which basis to find different conclusions, because: (A) We have not disputed that we would find the same experimental **results**; (B) our new assays give firm results *and* they are better tests of **interpretations**. If anything, it is better to ask whether, if the original papers had *also* performed the tests we have done, their interpretations would be in line with ours (case in point TRFS-green: the authors' answer is yes, **as we now discuss and highlight**).

Finally, we now discuss and highlight clearly the value of our new, expanded set of assays designed to probe cellular selectivity (see eg **Conclusions, p12 "...which we believe will be a valuable methods addition to the literature"**). In this way we make it clear that through our new experiments, we can test much better for selectivity, and when these experiments indicate a tool is selective, it is more likely to be of a quality that "can at last allow researchers to unveil the dynamics of these major dithiol/disulfide-type enzyme systems within cells." So we appreciate the advice, but we remain convinced our approach is solid.

Also, concluding their 2nd round review, the reviewer stated that we had not provided evidence to "overturn peer-reviewed results showing the efficacy of the TRFS". With repetition: there is no peer-reviewed result which shows a dithiolane probe is a cellular TrxR-selective reporter. There are peer-reviewed experimental results, and there are peer-reviewed interpretations of those results. We - and, following in our footsteps even the original authors - are overturning those peer-reviewed interpretations by provision of extra, new results: no overturning of a solid old scientific result is involved and none are needed.

Reviewer 3:

I congratulate the authors for an excellent and courageous paper.

We thank the reviewer for this evaluation. We too believe that there should be room in the literature for improving on the accuracy of previously-published interpretations, at a reasonable standard of methodology and proof; and we believe this paper is a step in the right direction that is worthy of publication.

=====

By addressing all issues raised in reviews, we believe our paper is now ready for publication. We confirm and highlighted that non-TrxR signal is kinetically competitive; we highlight Fang's recent TRFS-green re-evaluation work; and we have explained why repeating the more limited and less stringent assays previously done can neither verify nor deny our central thesis but the new experiments we provide are the key to correct scientific interpretations of the cellular TrxR-nonselectivity of dithiolane probes.